# Concerted catalysis of single atom and nanocluster enhances bio-ethanol activation and dehydrogenation

Zhao Sun [1,2], Weizhi Shi[1], Louise R. Smith [2], Nicholas F. Dummer [2], Haifeng Qi [2], Zhiqiang Sun [1] ✉ & Graham J. Hutchings [2] ✉

Single atom and nanocluster catalysts are extensively investigated in heterogeneous catalysis due to their high catalytic activity and atomic utilization, while their coexisting properties and potentially synergistic effect are yet to be clarified. Herein, we construct three systems of atomic-scale catalysts ($x$Ni/Mo$_2$TiAlC$_2$, $x = 0.5$, 1, and 1.5) for bio-ethanol reforming, which correspond to single atoms, single atoms mixed with nanoclusters, and nanoclusters. The respective hydrogen utilization efficiency of mixed-form catalyst increases by 43.7% and 29.3% compared to single atom and nanocluster catalysts. Results demonstrate that the adjacent Ni single atom facilitates electron transfer from Mo$_2$TiAlC$_2$ to Ni-Mo interface and raises the d-band center, thus enhancing bio-ethanol adsorption and activation; while the existence of Ni nanoclusters contributes to lowering the energy barriers of CH$_3$CHO* dehydrogenation. The catalytically active sites are Ni-Mo alloyed single atoms with adjacent Ni nanoclusters. This work provides new implications for highly activated catalytic site construction and advanced catalyst design.

Owing to its high gravimetric energy density and pollution-free products, hydrogen is considered one of the most promising alternatives to fossil fuels[1–4]. It can also be predicted that hydrogen energy will occupy an important position in the future energy landscape[5–7]. In this context, the steam reforming of bio-ethanol (SRE) plays an essential role in green hydrogen production[8–10]. Nickel is the benchmark metal for SRE due to its extraordinary capability for C-C and C-H bond dissociation in addition to its low cost[11,12]. However, Ni-based materials are prone to deactivation through sintering, agglomeration, and carbon deposition[13,14]. Our previous work utilized MXene as a carrier, loading 10 wt% Ni to form a highly stabilized Ni-Mo$_m$-Mo$_{2-m}$TiC$_2$T$_x$ structure, which enhanced metal-support interactions and inhibited catalyst deactivation[15]. Nevertheless, to the best of our knowledge, very few studies have explored ethanol reforming using a low-Ni-loading catalytic system. Reducing Ni loading while maximizing atom-utilization efficiency, even enhancing the catalytic performance, remains a significant yet challenging task.

Recently, single-atom catalysts (SACs) have garnered significant attention in photo-, electro-, and thermo-catalysis due to their exceptional atomic utilization efficiency and unique coordination environments, thereby enhancing reactivity and selectivity in catalytic processes[16–20]. Scaling down the size of Ni from nanoparticle to nanocluster and single atom leads to ultrahigh atom-utilization efficiency and enhances the interaction with the support[21–23]. For instance, Ni single atoms dispersed on Ce-doped hydroxyapatite and CeO$_2$ exhibits outstanding activity and resistance to coke formation during high-temperature dry reforming of methane[24,25]. However, while SACs benefit from highly dispersed and isolated active sites, the limited overall number compared to nanoparticles or nanoclusters can constrain their catalytic activity. This trade-off between atomic dispersion and site density poses a challenge for SACs in reactions requiring abundant active sites. In the case of ethanol reforming to hydrogen, the applicability and feasibility of Ni-based SACs remain unclear.

[1]Hunan Engineering Research Center of Clean and Low-Carbon Energy Technology, School of Energy Science and Engineering, Central South University, Changsha 410083, China. [2]Max Planck–Cardiff Centre on the Fundamentals of Heterogeneous Catalysis FUNCAT, Cardiff Catalysis Institute, School of Chemistry, Cardiff University, Cardiff CF24 4HQ, United Kingdom. ✉e-mail: zqsun@csu.edu.cn; Hutch@cardiff.ac.uk

Specifically, their effects on ethanol conversion and hydrogen selectivity require further exploration[26,27].

Atomic clusters are energetically favored and stable than single atoms under working conditions due to their lower surface free energy and the formation of strong metal-metal bonds. Previous studies have primarily focused on constructing either single-atom sites or cluster sites. Coupling nanoclusters (NCs) with single atoms (SAs) offers great potential for tailoring the electronic properties and coordination environments of the combined catalyst, thereby promoting catalytic activity or selectivity through a synergistic mechanism[28]. For example, the combination of $Ni_x$ NCs and Ni single atomic sites has been proved to perform a synergistic effect, improving $CO_2$ activation and optimizing the adsorption of key intermediates such as COOH*[29]. Similarly, Ma et al.[30] demonstrated that the proximity of Fe NCs to Fe single atoms facilitates molecular $O_2$ and amine substrate adsorption, promoting singlet oxygen generation and lowering the energy barrier for key intermediate imine formation via H-atom abstraction. Despite these advances, dual-metal catalytic systems remain complex. Challenges such as identifying the precise active sites and unraveling detailed reaction mechanisms hinder a deeper understanding of these systems. Addressing these intricate issues is essential for fully harnessing the potential of dual-metal species in catalysis[31].

In this work, we report the successful synthesis and catalytic performance of a series of $x$Ni/$Mo_2TiAlC_2$ catalysts (abbreviated as $x$Ni/ MTAC, $x$ = 0.5, 0.75, 1, 1.25, and 1.5), and further demonstrate the synergy of single atom and adjacent nanocluster. Specifically, we showed the synthesized 0.5Ni/MTAC, 1Ni/MTAC, and 1.5Ni/MTAC are primarily representative of single atoms, single atoms mixed with nanoclusters, and nanoclusters, respectively. Among them, 1Ni/MTAC catalyst exhibits outstanding SRE activity with a hydrogen utilization efficiency (HUE) of 67.8% at 550 °C and remains stable without obvious deactivation over 120 h. Compared with our previous work on the structural regulation of MXene-loaded 10 wt% Ni nanoparticles, the hydrogen yield per Ni atom increased by 700%[15]. We reveal that the adjacent Ni single atoms of NCs can promote the ethanol adsorption and activation, while that of Ni nanoclusters play the role in lowering the energy barrier of the rate-determining step of $CH_3CHO$* dehydrogenation.

## Results

### Characterization of Ni/MTAC catalyst

The fabrication process of the Ni/MTAC catalysts is illustrated schematically in Fig. 1a. The Ni/MTAC catalysts were prepared by an incipient-wetness impregnation method. Ni was impregnated over the $Mo_2TiAlC_2$ support with loadings of 0.50 wt%, 0.75 wt%, 1.00 wt%, 1.25 wt%, and 1.50 wt%, denoted as $x$/MTAC (x = 0.5, 0.75, 1, 1.25, and 1.5). For comparison, Ni catalysts with 1 wt% loading were prepared by the same method over $Mo_2TiC_2$ (MXene), $Al_2O_3$, $SiO_2$, and $CeO_2$ supports, denoted as 1Ni/MTC, 1Ni/$Al_2O_3$, 1Ni/$SiO_2$, and 1Ni/$CeO_2$, respectively. Synthetic details are provided in the Methods section.

The $N_2$ adsorption and desorption isotherms were performed to determine the specific surface area and pore structure parameters of the as-prepared samples (Supplementary Fig. 1 and Supplementary Table 1). The BET surface areas of the catalyst with other supports were similar to the Ni/MTAC catalysts except for 1Ni/$Al_2O_3$. The actual Ni metal loadings of the as prepared catalysts were measured by inductivity-coupled plasma atomic emission spectroscopy (ICP-AES), and the results were close to the nominal values (Supplementary Table 2).

The X-ray diffraction patterns (XRD) of all the as prepared Ni/ MTAC catalysts evidenced no crystalline Ni or NiO phase (Fig. 1b), suggesting that Ni atoms were highly dispersed on the MTAC support. The diffraction peaks at 9.5° and 19.1° correspond to the crystal planes of (002) and (004), respectively, which are consistent with the pure MTAC support[32], confirming its crystalline structure remained intact.

Scanning electron microscopy (SEM), transmission electron microscopy (TEM), and high-angle annular dark-field scanning transmission electron microscopy (HAADF-STEM) measurements were carried out to analyze the morphology and structure of the catalysts. The SEM image of the 1Ni/MTAC reveals the layered structure of MTAC (Fig. 1c), with similar results for other Ni-loaded samples (Supplementary Fig. 2). Likewise, the TEM image further confirms the layered structure of the 1Ni/MTAC, which features smooth surfaces and edges (Fig. 1d). According to previous reports[33], the structure of MTAC is that Ti atoms are sandwiched between two Mo-layers, and the Mo layer is adjacent to the Al planes, resulting in a Mo-Ti-Mo-Al-Mo-Ti-Mo stacking order (Fig. 1e). The C atoms are located between the Mo and Ti layers.

For as prepared 0.5Ni/MTAC sample, the HAADF-STEM image showed numerous distinct bright dots that can be distinguished beyond the MTAC surface, demonstrating the primary existence of Ni single atom along with the EELS results (~89% frequency, Fig. 1f and Supplementary Figs. 3, 4). For 1Ni/MTAC, relatively low magnification image showed the presence of ~0.9 nm nanoclusters (Fig. 1g and Supplementary Fig. 5), whereas higher magnification images clearly revealed the individual Ni atoms distributed over the MTAC support (Fig. 1h). Moreover, well-resolved lattice fringes with a d-spacing of 0.467 nm were observed, which corresponds to the MTAC(004) plane, confirming that the 1Ni/MTAC is comprised of both Ni single atom (~38% frequency) and nanoclusters (~62% frequency). In contrast, Ni NCs are the predominate structure on the 1.5Ni/MTAC (~85% frequency). Meanwhile, the energy dispersive X-ray spectroscopy (EDS) element mapping results exhibited the homogeneous distribution of Ni over the surface of 1Ni/MTAC (Fig. 1j and Supplementary Fig. 6).

X-ray photoelectron spectroscopy (XPS) analysis of $x$Ni/MTAC revealed the coexistence of $Ni^{\delta+}$ (852.4 → 853.0 eV, 0 < $\delta$ < 2) and $Ni^{2+}$ (855.4 → 855.8 eV) species. The $Ni^{\delta+}$ species correspond to Ni nanoclusters, while the $Ni^{2+}$ species, exhibiting a higher oxidation stage, are attributed to Ni single atoms due to the electronic structure modifications induced by their atomic dispersion (Fig. 2a and Supplementary Fig. 7)[34,35]. As the Ni content decreases from 1.5 to 0.5 wt%, the proportion of $Ni^{\delta+}$ species reduces, accompanied by a gradual shift of its peaks to higher binding energies (Supplementary Table 3). The shift of the $Ni^{\delta+}$ peak towards higher binding energies with Ni contents is possibly attributed to the reduction in cluster size, which leads to a change in its coordination environment, resulting in a decrease in the electron cloud density around the Ni clusters.

Additionally, the Mo 3$d$ spectrum was deconvoluted into four distinct peaks at 227.5, 228.8, 231.9, and 235.3 eV, corresponding to metallic Mo, Mo-C ($3d_{5/2}$), Mo-C ($3d_{3/2}$), and Mo-$O_x$ bonds, respectively (Fig. 2b). The presence of Mo-C and Mo-O bonds is further corroborated by the Raman spectra (Supplementary Fig. 8). Moreover, high-resolution XPS spectra of Ti 2$p$, C 1$s$, Al 2$p$, and O 1$s$ provide additional confirmation of the layered structure of MTAC (Supplementary Figs. 9–13).

The electronic structure and coordination environment of Ni and Mo species was further determined by X-ray absorption spectroscopy (XAS) measurements. The normalized X-ray near-edge absorption spectra (XANES) of Ni K-edge are presented in Fig. 2c. The absorption edges of three samples were all located between Ni foil and NiO, following the valence order of 0.5Ni/MTAC > 1Ni/MTAC > 1.5Ni/MTAC. This suggests the positive charge of the Ni atoms, which is in agreement with the XPS results[36]. Besides, the adsorption edge of 1.5Ni/ MTAC shows the smallest shift to the higher energy of the three catalysts, indicative of the lowest electron density of interfacial Ni atom. As shown in normalized XANES spectra of Mo K-edge (Fig. 2d), the absorption edge of all samples is located between the Mo foil and $MoO_2$, suggesting the positively charged state of Mo, which is in good agreement with the XPS results.

The coordination environments were further analyzed by the extended X-ray absorption fine structure (EXAFS) at the K-edge of Ni

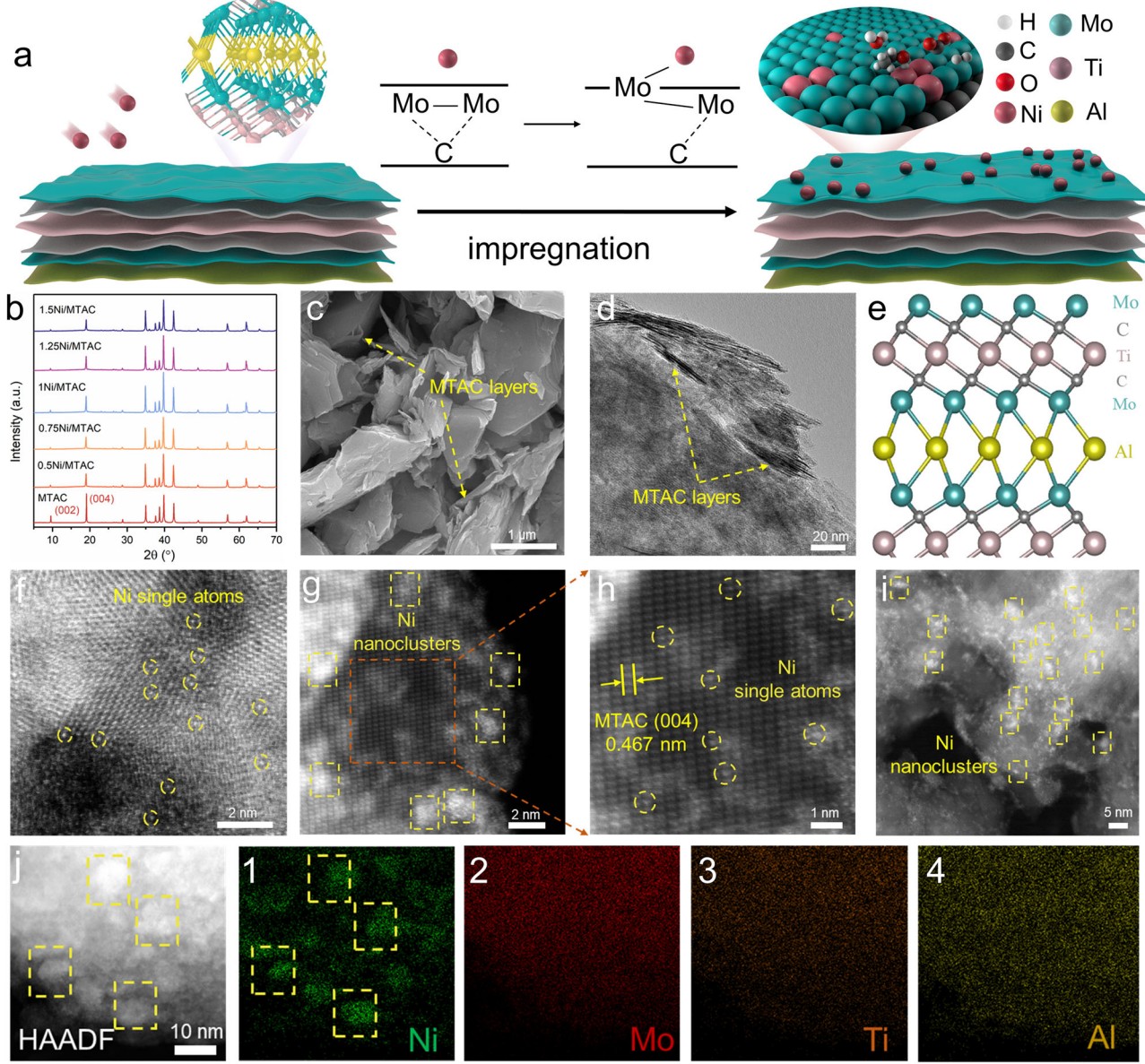

**Fig. 1 | Synthesis strategy and structural characterization of the fabricated Ni/ MTAC catalysts. a** schematic illustration of the SRE with applied as prepared Ni/ MTAC catalysts for renewable hydrogen production. **b** XRD patterns of the as prepared Ni/MTAC catalysts. **c** SEM image, and (**d**) TEM image show the few layers of 1Ni/MTAC. **e** Depth profiles of Mo$_2$TiAlC$_2$ sample. **f** HAADF-STEM image of 0.5Ni/ MTAC. **g, h** Magnified HAADF-STEM images of 1Ni/MTAC. **i** HAADF-STEM image of 1.5Ni/MTAC. **j** HAADF-STEM image and corresponding EDS maps of 1Ni/MTAC. The yellow squares indicate Ni nanoclusters and some of the single Ni atom sites are highlighted by yellow circles.

and Mo. The Ni K-edge EXAFS of Ni foil shows a main peak at 2.09 Å (not phase-corrected), which was attributed to the Ni-Ni scattering path (Fig. 2e and Supplementary Figs. 14, 15). With the decrease in the Ni loadings, the peak gradually shifted to longer R value, owing to the formation of Ni-Mo path. Compared to Ni foil, the intensity of the Ni-Ni scattering path gradually diminished, confirming the interaction between Ni and MTAC support with the formation of Ni-Mo path, corresponding to coordination number reduction from 12.0 to 9.6, 8.5, and 6.7 (Supplementary Table 4). The lengths of Ni-Ni/Mo bonds in 0.5, 1, and 1.5Ni/MTAC further increased compared to Ni foil, testifying the formation of Ni-Mo bonds. Moreover, with the increase of Ni content, the coordination number of Ni-Mo decreases, resulting in the weakening of metal-support interaction.

For Mo-edge EXAFS, all the Ni/MTAC samples exhibited strong Mo-C, Mo-Ni, and Mo-Mo scattering (Fig. 2f and Supplementary Figs. 16, 17)[37]. The 2.27 Å contribution (not phase-corrected) was

attributed to the Mo-Ni scattering, which is slightly shifted compared with the Mo-Mo scattering in Mo foil (Supplementary Table 5)[38]. To precisely clarify the atomic dispersion and coordination conditions of Ni and Mo, the wavelet transform of EXAFS spectra were analysed (Supplementary Figs. 18, 19), further confirming the formation of the Ni-Mo coordination.

## Catalytic SRE performance

The effects of different Ni loadings (0.50, 0.75, 1.00, 1.25, and 1.50 wt%) on the gas composition and hydrogen utilization efficiency (HUE) were investigated (Fig. 3a and Supplementary Figs. 20–23). A trace amount of ethane and ethylene was detected under 0.5Ni/MTAC and 0.75Ni/ MTAC catalysts, attributed to their weaker catalytic activity in C-C bond cleavage. Simultaneously, the existence of Mo species could promote the formation of ethane via ethanol dehydrogenation and ethylene hydrogenation, leading to the production of C$_2$ products at

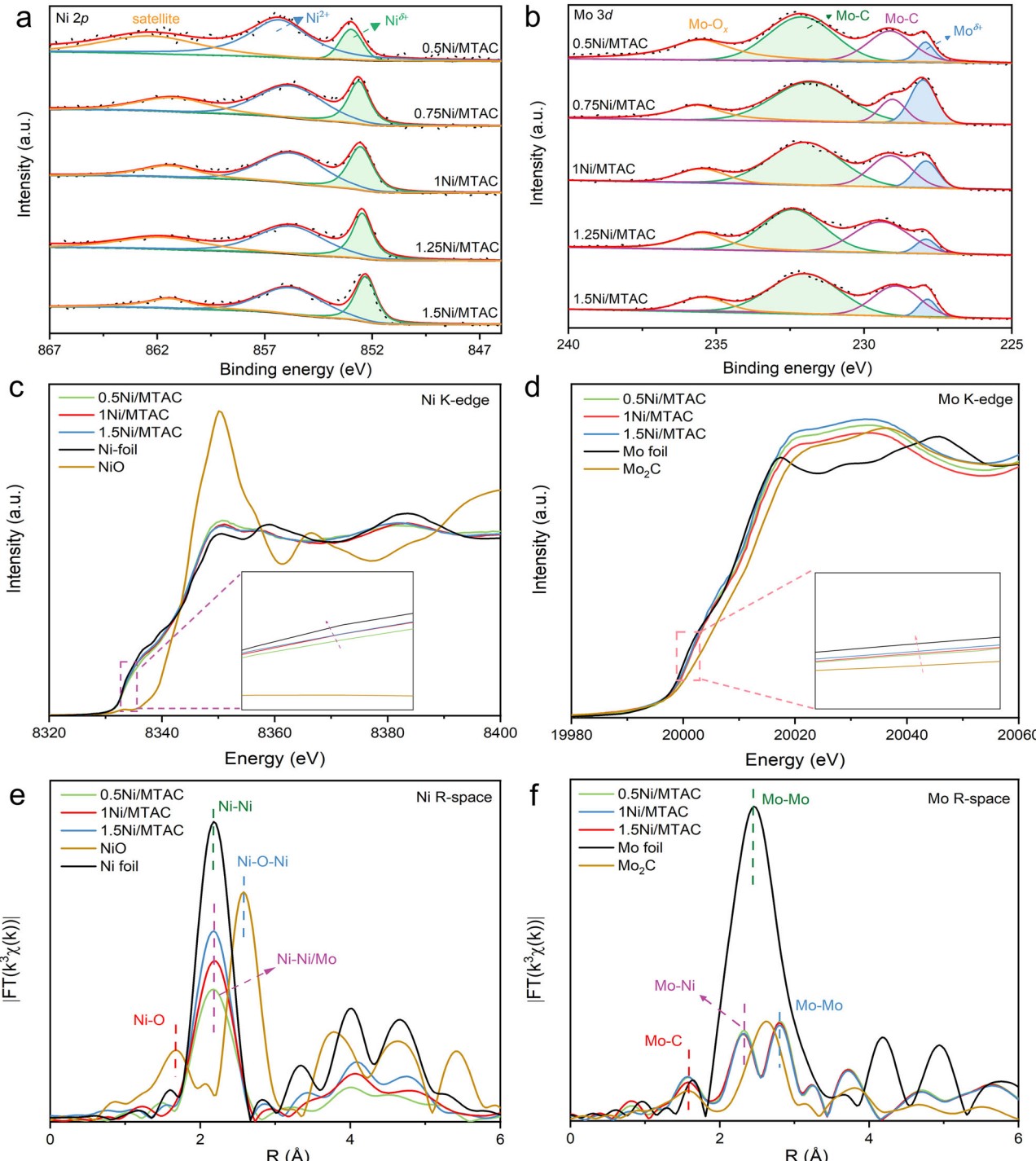

**Fig. 2 | Electronic state and atomic structure characterization. a** High resolution Ni 2*p* spectra and (**b**) Mo 3*d* spectra. **c** The normalized Ni K-edge XANES spectra. **d** The normalized Mo K-edge XANES spectra. **e** Corresponding Fourier-transformed R-space spectra (solid lines) and fits (open points) of Fig. 2c. **f** Corresponding Fourier-transformed R-space spectra of Fig. 2d.

low Ni/Mo ratios. The maximum HUE was present in the 1Ni/MTAC sample, implying comprehensively optimal ethanol conversion and $H_2$ selectivity. Specifically, increasing the Ni loading from 0.50 to 0.75 and 1.00 wt% can remarkably improve HUE from 24.1% to 41.5% and 67.8%, respectively. Ethane and ethylene by-products completely vanished over the 1Ni/MTAC catalyst, which is consistent with the results of ethanol pulse experiments (Supplementary Fig. 24). However, upon further increasing Ni loadings from 1.00 to 1.25 and 1.50 wt%, the HUE of 1.25Ni/MTAC and 1.5Ni/MTAC decreased by 13.8% and 29.3%,

respectively. This sufficiently demonstrates the superiorities of concerted catalysis of Ni single atom and Ni nanoclusters, which improve the catalytic performance of SRE. We estimated the number of active sites of 0.5Ni/MTAC, 1Ni/MTAC, and 1.5Ni/MTAC catalysts under a same mass (Supplementary Fig. 25). This, to some extent, also reveals the reason for the high activity of the 1Ni/MTAC sample.

Additionally, the effects of reaction temperatures and gaseous hourly space velocity were also explored with the 1Ni/MTAC sample. Higher temperatures (>550 °C) were not able to further improve HUE

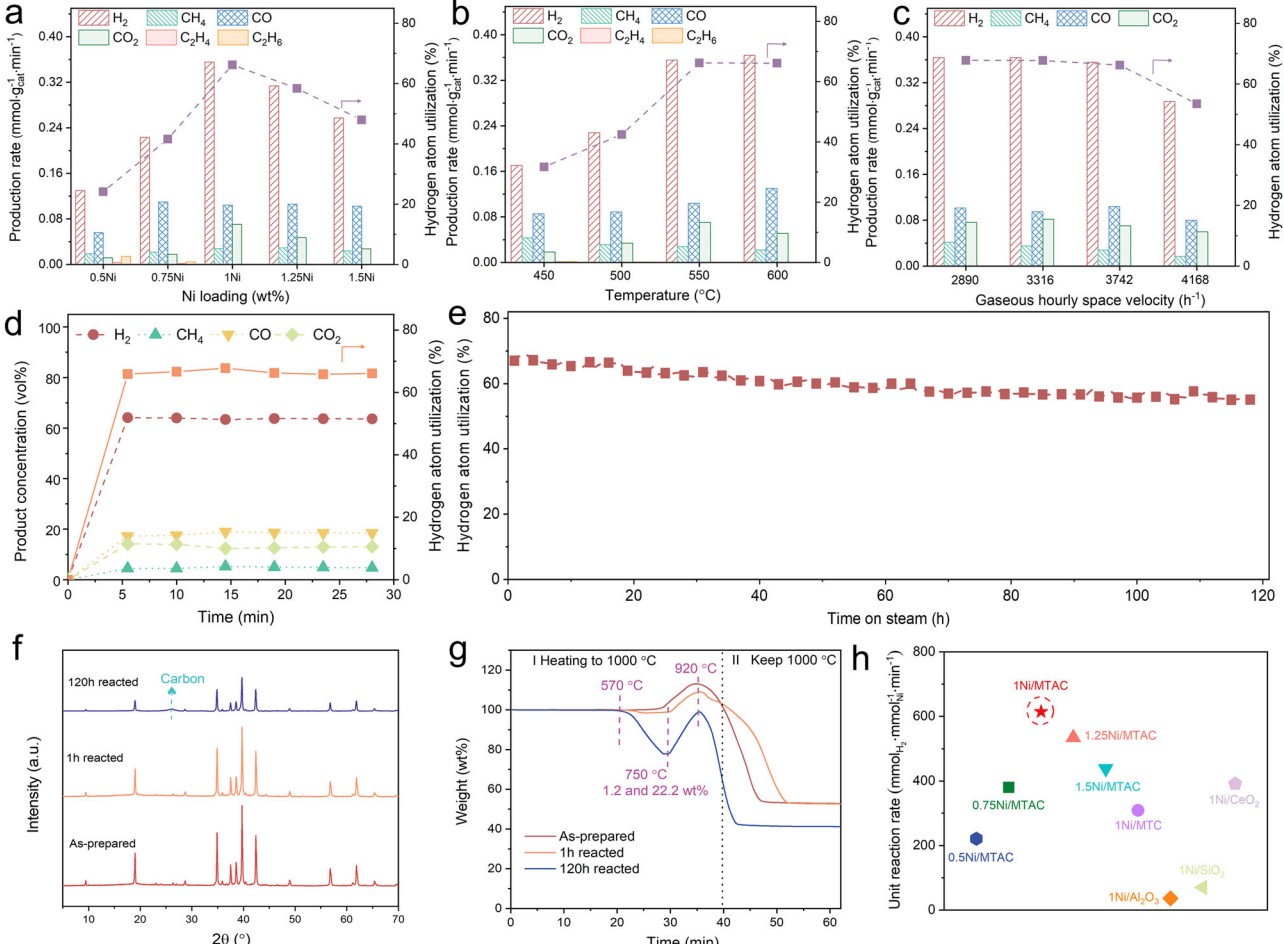

**Fig. 3 | SRE performances of all the as prepared catalysts.** Effect of (**a**) Ni loadings, (**b**) reaction temperatures, and (**c**) gaseous hourly space velocity on gas concentration and hydrogen utilization efficiency at 20 min of SRE. **d** Product concentration and hydrogen utilization efficiency as a function of time using the 1Ni/MTAC catalyst. Activity test conditions: 1 atm, 550 °C, S/E = 3, liquid feeding rate 0.005 mL min⁻¹, N₂ flow rate 40 mL min⁻¹. **e** Time-on-line hydrogen utilization efficiency over 1Ni/MTAC. **f** The XRD pattern of the as prepared and reacted 1Ni/MTAC catalysts. **g** TGA profiles of 1Ni/MTAC catalysts after different reaction times on-stream at 550 °C. **h** Comparison of unit hydrogen yield of the Ni supported by different substrates.

(Fig. 3b and Supplementary Figs. 26-28). The increased CO yield was due to the enhancement of the reverse water-gas shift reaction (RWGS), with a corresponding decrease in the concentration of CO₂. Notably, 1Ni/MTAC showed a good catalytic performance with HUE values of 42.5% and 31.8% at 500 °C and 450 °C, respectively. Moreover, the concentrations of by-products ethane and ethylene were lower than 0.2 vol% and 0.3 vol%, respectively. The results above indicate the potential of 1Ni/MTAC sample for low-temperature SRE. The HUE reached 70% under a GHSV no more than 3742 h⁻¹ (Fig. 3c and Supplementary Figs. 29–32). When the GHSV increased to 4168 h⁻¹, the HUE still remains 53.5%. As shown in Fig. 3e, the 1Ni/MTAC catalyst show satisfied long-term durability on catalytic SRE, as its HUE remains >55% after 120 h of SRE. Moreover, compared with the 10Ni/MTAC catalyst in our previous studies[15], the catalytic activity per unit Ni atom unexpectedly increased by 700%, demonstrating the superiority of 1Ni/MTAC catalysts (Supplementary Fig. 33).

Compared with the as prepared sample, the characteristic diffraction patterns of samples reacted for 1 h and 120 h remained basically unchanged, with no Ni or NiO peak being observed (Fig. 3f), suggesting the remarkable resistance of 1Ni/MTAC to sintering and agglomeration. Additionally, a characteristic peak of carbon was also observed after 120 h long-term reaction, indicating that the carbon deposition originating from ethanol, as the MTAC phase remained stable. Even after extended testing (120 h), the morphology and the

structure of the catalyst were still well maintained. Nevertheless, the formation of carbon nanotubes (CNTs) was also observed (Supplementary Fig. 34), which is consistent with the XRD results. The amount of carbon formed on the surface of the catalyst was studied by Thermogravimetrc analysis (TGA), as shown in Fig. 3g. The weight loss of the 120 h reacted sample started at 570 °C and was attributed to the oxidation of the carbon nanotubes. When the temperature reached 750 °C, metallic Ni and Mo are oxidized, leading to the weight increase observed for all the catalysts. With the temperature further increased (>920 °C), the weight loss for all the samples (as prepared and reacted catalysts) are basically identical, which is assigned to the oxidation of carbon inside MTAC. This result preliminary deduced that the carbon nanotubes come from ethanol rather than the support MTAC, due to the thermal stability of the as-prepared sample.

In comparison with 1Ni/MTAC, the HUE over the other 1Ni catalysts with different supports was significantly decreased (Fig. 3h), revealing the benefit of using MTAC as a catalyst support. Specifically, the 1Ni/MTC and 1Ni/SiO₂ showed higher concentrations of CO, and the 1Ni/Al₂O₃ catalyst displayed higher selectivity to ethylene rather than hydrogen (Supplementary Figs. 35–37). Although 1Ni/CeO₂ achieved a higher initial hydrogen yield, we attribute this to the promotion of lattice oxygen within the CeO₂ (Supplementary Fig. 38). As the consumption of lattice oxygen progresses, the SRE performance decreased accordingly. Comparatively speaking, the above results

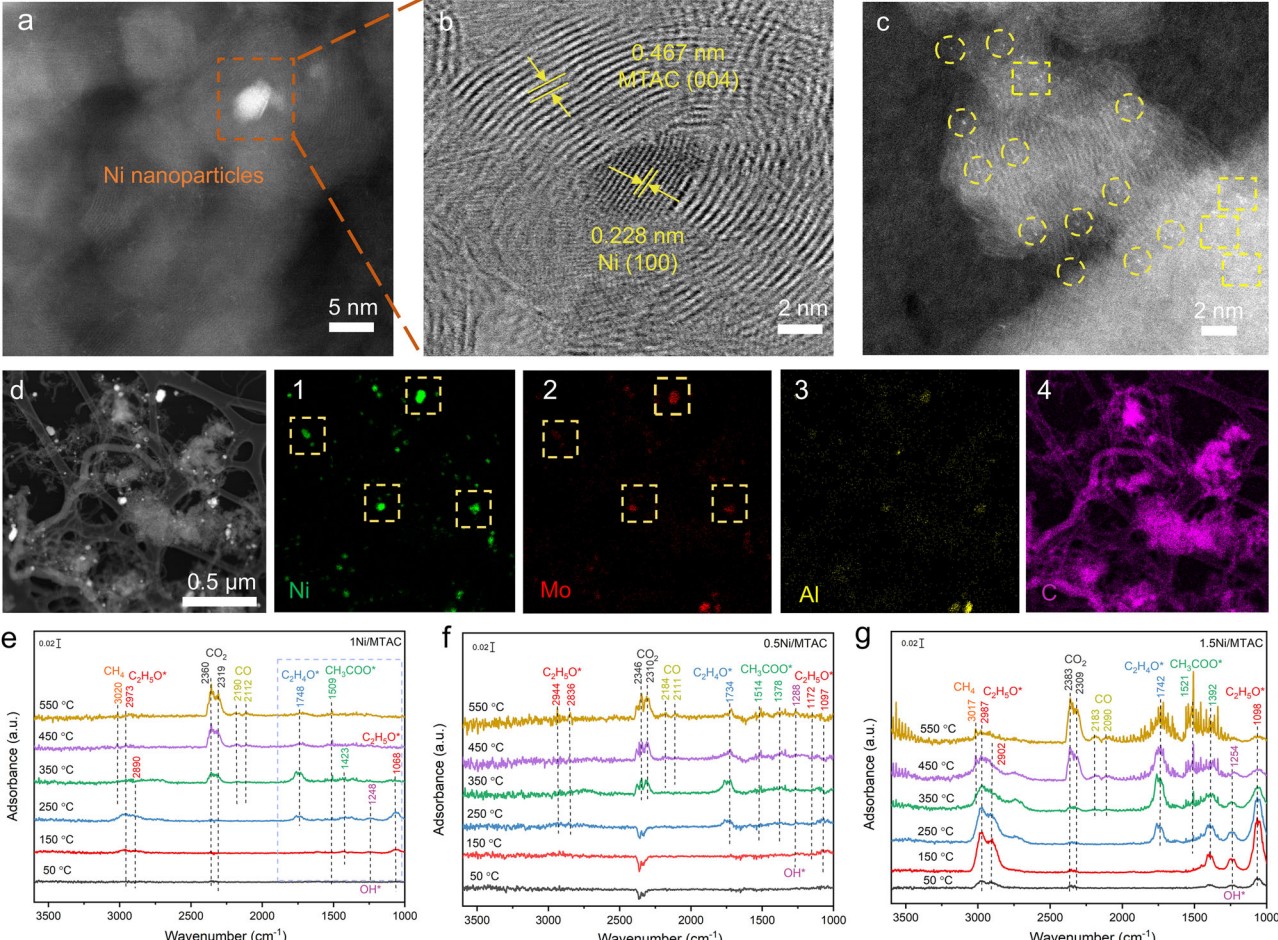

**Fig. 4 | Morphology characterization of 120 h reacted 1Ni/MTAC and the results of In situ DRIFTS. a** HAADF-STEM image, and (**b, c**) magnified HAADF-STEM image of the 120 h reacted 1Ni/MTAC. **d** Corresponding EDS maps of the 120 h reacted 1Ni/ MTAC. The yellow squares indicate Ni nanoclusters and some of the single Ni atom sites are highlighted by yellow circles. In situ DRIFTS of (**e**) 1Ni/MTAC, (**f**) 0.5Ni/ MTAC, and (**g**) 1.5Ni/MTAC.

demonstrate that the atomically dispersed Ni over the MTAC surface are remarkable on catalytic SRE (Supplementary Table 6). Significantly, the Mo atoms of MTAC trap isolated Ni atoms, which efficaciously prevent on-stream sintering and agglomeration, such that the catalyst is highly resistant to catalyst deactivation. Besides, the synergy of a single atom and nanocluster was adequately testified.

## SRE mechanism clarification

The 120 h reacted 1Ni/MTAC sample was therefore examined by HAADF-STEM (Fig. 4a). The morphology and the structure of the catalyst were still well retained (Fig. 4b). In addition, the magnified HAADF-STEM image also identified that the Ni single atoms and NCs still co-exist (Fig. 4c). After the 120 h long-term reaction, the distributions of Ni and Mo elements were highly consistent (Fig. 4d), which is in agreement with the XPS results (Supplementary Fig. 39), further supporting the formation of the Ni-$Mo_n$-$Mo_{2-n}$TiAlC$_2$ structure.

In situ diffuse reflectance infrared Fourier transform spectroscopy (in situ DRIFTS) measurements were conducted to elucidate the ethanol evolution pathway under the induction of 1Ni/MTAC (Fig. 4e). The bond scission of ethanol over metallic Ni has been previously reported, and generally follows the sequence: O-H bond → C-H bond in methylene (-$CH_2$-) → C-C bond → C-H bond in methyl (-$CH_3$)[39]. When the ethanol-water mixture was introduced to the surface of 1Ni/MTAC, the C-O stretching bands of ethoxy ($CH_3CH_2O^*$) species were observed, including a bidentate-$v$(CO) stretch at 1068 cm$^{-1}$. Respectively, the C-H vibrations peaks of ethoxy at 2890 and 2973 cm$^{-1}$ could be assigned to

$v_{as}(CH_3)$ and $v_s(CH_3)$ vibrations[40,41]. These features indicate that the ethoxy group was formed first after the cleavage of the O-H bond of adsorbed ethanol. The broad hump at 3000-3600 cm$^{-1}$ can be assigned to the hydroxyl groups of the adsorbed water[42], consistent with the $H_2$-TPR results (Supplementary Fig. 40). As the temperature reached 150 °C, the peaks of $v_{as}$(OCO) at 1509 cm$^{-1}$ and $\delta_s(CH_3)$ at 1423 cm$^{-1}$ were detected, the intensity of which increased with the rise of the temperature. This signifies the formation of acetate ($CH_3COO^*$) intermediate[43]. In addition, the characteristic aldehyde (HCO*) peak emerged at 1748 cm$^{-1}$ and the band of $\delta$(OH) in ethanol at 1248 cm$^{-1}$ was not detected (350 °C), implying the formation of acetaldehyde at elevated temperature[44]. These changes support the inference that the primary and important intermediates for SRE are ethoxy, acetaldehyde, and acetate species. At higher temperatures, the acetate species decomposed and was accompanied by the release of $CO_2$ and methane, as confirmed by the peaks in the range of 2310–2380 cm$^{-1}$ and the peak at 3020 cm$^{-1}$.

For comparison, the performance of 0.5Ni/MTAC, 1.5Ni/MTAC (Fig. 4f, g), and other 1Ni catalysts with different supports ($Al_2O_3$, MTC, $SiO_2$, $CeO_2$) were also studied (Supplementary Figs. 41–45), which helped elucidate the superior performance of MTAC supported Ni-based catalysts in SRE process. At higher temperatures (>350 °C), the intensity of the ethoxy signal peaks on other catalysts is significantly higher than that of 1Ni/MTAC, suggesting that the 1Ni/MTAC catalyst can promote the transformation of ethoxy to acetate. Moreover, the $\delta$(OH) band in ethanol at 1248 cm$^{-1}$ could also be observed on other

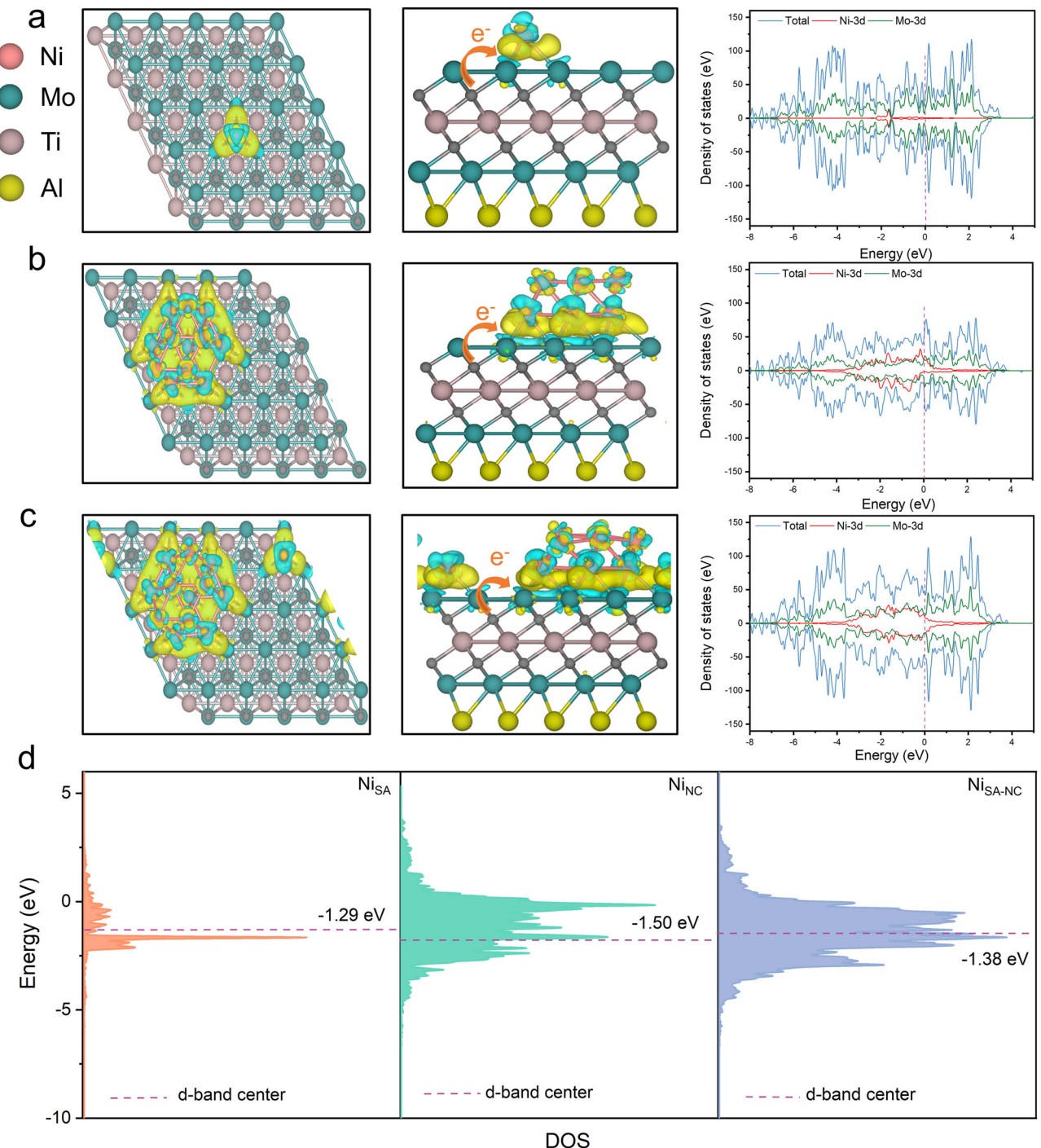

**Fig. 5 | Analysis of bader charge and DOS.** Structure representations of (**a**) $Ni_{SA}$/ MTAC(004), (**b**) $N_{NC}$/MTAC(004), and (**c**) $Ni_{SA-NC}$/MTAC(004) as well as their calculated bader charge and DOS of surface models. **d** The d-band centres of different nanostructured architectures on the surface of MTAC(004). The $Ni_{SA}$ model with one Ni atom, the $Ni_{NC}$ model with 12 Ni atoms, and the NiSA-NC model, which features one Ni atom adjacent to 12 Ni atoms.

catalysts, indicating an inferior ability for activating absorbed ethanol. The peak of the acetaldehyde intermediate was also detected on the other catalysts, with the presence of acetaldehyde predominately favoring the generation of CO and methane at elevated temperatures[45], which is in agreement with the observed SRE performance.

## DFT calculations
On the basis of the HAADF-STEM results, Bader charge, density of states (DOS), and energy levels of d-band centre of $Ni_{SA}$/MTAC(004), $Ni_{NC}$/MTAC(004), and $Ni_{SA-NC}$/MTAC(004) were calculated to study

the interplay between the single atom and the adjacent nanoclusters (Fig. 5). The catalytic system containing Ni single atoms (Fig. 5a, c) shows promoted charge transferability from the MTAC support to the Ni-Mo bimetallic interface compared with $Ni_{NC}$/MTAC(004) (Fig. 5b), which is consistent with the XAS results. Specifically, the electron transfer amount follows the sequence of $Ni_{SA}$/MTAC(004) > $Ni_{SA-NC}$/ MTAC(004) > $Ni_{NC}$/MTAC(004), as shown in Table 1. These results indicated that the combination of two catalytic forms affects electron transmission by interacting with the substrate, thus enhancing the catalytic performance.

The DOS of the surface near the Fermi level mainly originate from the Ni $3d$ and Mo $3d$ states. Moreover, as shown in Fig. 5d, the calculated d-band center energy levels for $Ni_{SA}$/MTAC(004), $Ni_{NC}$/MTAC(004), and $Ni_{SA-NC}$/MTAC(004) are -1.29, -1.50, and -1.38 eV, respectively. This shows that the d-states of the catalysts with the existence of single atoms ($Ni_{SA}$/MTAC and $Ni_{SA-NC}$/MTAC) are closer to the Fermi level, which is beneficial to enhancing the adsorption of intermediates during the SRE process.

Underpinned by the O-H bond activation, several key steps in $CH_3CH_2OH$ dehydrogenation ($CH_3CH_2OH^* \rightarrow CH_3CH_2O^* \rightarrow CH_3CHO^* \rightarrow CH_3CO^*$) over $Ni_{SA}$/MTAC(004), $Ni_{NC}$/MTAC(004), and $Ni_{SA-NC}$/MTAC(004) were calculated (Fig. 6 and Supplementary Figs. 46, 47). For the first O-H bond cleavage, the reaction energy barriers over $Ni_{NC}$/MTAC(004) and $Ni_{SA-NC}$/MTAC(004) are 0.59 and 0.58 eV, respectively. These two barriers are considerably lower than that over $Ni_{SA}$/MTAC(004) (1.08 eV), implying the enhancement of Ni nanoclusters in

O-H bond activation. For the next methylene dehydrogenation step, the energy barrier of $Ni_{SA-NC}$/MTAC(004) decrease by 0.15 and 0.45 eV compared to $Ni_{SA}$/MTAC(004) and $Ni_{NC}$/MTAC(004), respectively, demonstrating the overwhelming superiority and synergistic effect of single atoms and nanoclusters towards ethanol dehydrogenation.

According to the calculations, dehydrogenation of acetaldehyde is the rate-determining step with energy barriers of 2.97, 1.64, and 1.28 eV on $Ni_{SA}$/MTAC(004), $Ni_{NC}$/MTAC(004), and $Ni_{SA-NC}$/MTAC(004), respectively. This can be attributed to the stronger adsorption of $CH_3CHO^*$ on the single atom sites, which is consistent with the in situ DRIFTS and DOS results. Overall, the synergy of single atoms and nanoclusters results in a moderate adsorption of the intermediate species and sharply decreased energy barriers for intermediates dehydrogenation on $Ni_{SA-NC}$/MTAC(004), thus boosting the catalytic activity of bio-ethanol reforming.

When two Ni single atoms ($Ni_{SA-SA}$) and two Ni nanoclusters ($Ni_{NC-NC}$) are introduced, the energy barriers for the transition states remain basically similar tendencies, i.e., the computational results align with the previous conclusions (Supplementary Figs. 48, 49). This confirms that the adsorption of the intermediate on the supported Ni single atom is still strong, while the introduction of nanocluster modulates an appropriate adsorption for ethanol activation as well as lower the energy barriers for the dehydrogenation of ethanol and other intermediates. Specifically, the barrier of 1.5Ni/MTAC for $C_2H_5O^*$

### Table 1 | Bader charge analysis

| Surface model | Total charge density (eV) |
|---|---|
| $Ni_{SA}$/MTAC | 0.27 |
| $Ni_{NC}$/MTAC | 0.13 |
| $Ni_{SA-NC}$/MTAC | 0.14 |

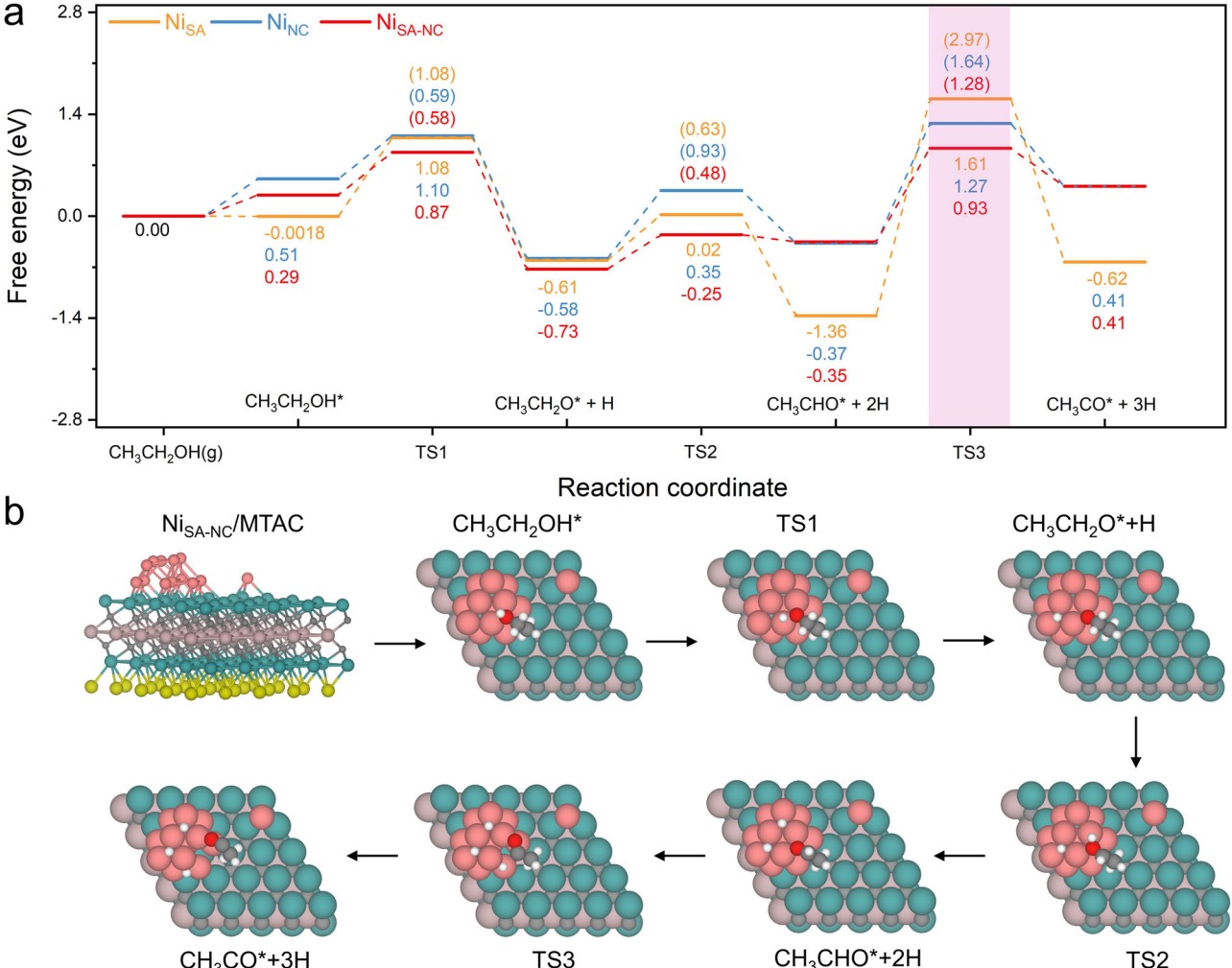

**Fig. 6 | DFT calculations of $CH_3CH_2OH$ dehydrogenation. a** Potential diagram from DFT calculations of $CH_3CH_2OH$ dehydrogenation over $Ni_{SA}$/MTAC(004) (orange), $Ni_{NC}$/MTAC(004) (blue), and $Ni_{SA-NC}$/MTAC(004) (red). **b** corresponding geometries over $Ni_{SA-NC}$/MTAC(004). Numbers in parentheses indicate the activation barriers for elementary steps in eV.

hydrogenation is still high, this is in agreement with the in situ DRITFS results. Additionally, the rate-determining step of all catalyst remains the dehydrogenation of acetaldehyde, with the transition state barrier of 0.5Ni/MTAC and 1.5Ni/MTAC decreasing from 2.97 eV to 2.89 eV and 1.64 eV to 1.54 eV.

## Discussion

In summary, we have constructed three types of Ni-supported catalysts, which are Ni single atoms, Ni nanoclusters, as well as a mix of both structures, for boosted bio-ethanol reforming. Among these catalysts, 1Ni/MTAC exhibited the highest catalytic performance, with an increase in hydrogen yield per Ni atom of 700% compared with 10Ni/MTAC. Moreover, the HUE was relatively stable after 120 h long-term SRE, attributing to the formation of Ni-Mo coordination and the Ni-$Mo_n$-$Mo_{2-n}$TiAlC$_2$ structure. Characterization results revealed the synergy of single atoms and nanoclusters, with enhanced electronic transfer and conversion of intermediates due to the concerted catalysis of the mixed Ni structure. DFT calculations further substantiate that the 1Ni/MTAC catalyst is more effective for facilitating the activation of ethanol and the subsequent dehydrogenation of the intermediate species. The coexistence of Ni single atom and nanoclusters markedly facilitated the rate-determining step of $CH_3CHO^*$ dehydrogenation. These findings provide new inspirations for the design and development of highly activate and stable catalysts with mixed structures, which will substantially promote the application of bio-ethanol reforming technologies with high hydrogen selectivity and yield.

## Methods

### Synthesis of Ni/MTAC catalyst

The Ni/MTAC catalysts were prepared via the incipient impregnation method. Briefly, a calculated amount of nickel nitrate hexahydrate ($Ni(NO_3)_2 \cdot 6H_2O$) was dissolved in the appropriate amount of DI water to obtain a solution of 0.05 g Ni mL$^{-1}$. Subsequently, the formulated solution was gradually dropped onto the MTAC support until saturation. A range of nominal Ni loadings (0.5, 0.75, 1, 1.25, and 1.5 wt%) were impregnated by changing the concentration of the $Ni(NO_3)_2$ solution. The resulting impregnated powders were dried in a vacuum drying oven at 60 °C for 12 h. Prior to the catalytic reaction, the obtained precursors were treated in 20 vol% $H_2$ in $N_2$ (50 mL min$^{-1}$) at 600 °C for 2 h. The resulting samples with various Ni content were denoted as 0.5Ni/MTAC, 0.75Ni/MTAC, 1Ni/MTAC, 1.25Ni/MTAC, and 1.5Ni/MTAC.

According to the same method described above, 1 wt% Ni loaded MXene, $Al_2O_3$, $SiO_2$, $CeO_2$ catalysts, were prepared by conducting the MXene ($Mo_2TiC_2T_x$, prepared by the previous HF etching method), commercial aluminum oxide ($Al_2O_3$), commercial silicon oxide ($SiO_2$), and commercial cerium oxide ($CeO_2$), as the supporting substrates. The obtained samples were named as 1Ni/MTC, 1Ni/$Al_2O_3$, 1Ni/$SiO_2$, and 1Ni/$CeO_2$, respectively.

### Materials Characterization

Actual Ni loadings were determined by inductively coupled plasma atomic emission spectrometry (ICP-AES, PerkinElmer Avio500).

The powder X-ray diffraction (XRD) patterns were recorded on a Advance D8 diffractometer using a Cu Kα radiation (λ = 1.5406 Å, 40 kV, and 40 mA) with a step size of 0.02° and a scanning rate of 7° min$^{-1}$. Patterns were collected in a range of 5°-70°.

BET surface areas were measured by $N_2$ porosimetry on a Micrometric TriStar 3000 analyzer with adsorption-desorption isotherms recorded at -196 °C. Before analysis, the catalysts were outgassed at 573 K for 3 h. Surface areas were calculated from the adsorption branch using the Brunauer-Emmett-Teller (BET) method. The pore size and pore volume of the Ni/MTAC catalysts were obtained via the Barret-Joyner-Halenda (BJH) method.

$H_2$ temperature-programmed reduction ($H_2$-TPR) measurements were performed on a Micromeritics AutoChem II 2920 instrument. For each experiment, 0.100 g sample was first put into a U-shaped quartz reactor, preheated at 300 °C in Ar (30 mL min$^{-1}$) for 30 min. The sample was then cooled to 50 °C followed by flowing 10% $H_2$ in Ar (30 mL min$^{-1}$) over the surface for 60 min. After that, the temperature was ramped from 30 °C to 1050 °C with a heating rate of 10 °C min$^{-1}$, and the results were recorded by the TCD detector.

Ethanol pulse reactions were also measured on the same chemisorption apparatus (Micromeritics AutoChem II 2920). The 0.100 g sample was purged under Ar (30 mL/min) flow from room temperature to 550 °C at a rate of 10 °C min$^{-1}$. Then, pulses of ethanol (0.50 mL, 2 min apart) were added to the reactor. After each pulse, The $H_2$ (m/z = 2), $CH_4$ (m/z = 16), CO (m/z = 28), and $CO_2$ (m/z = 44) were recorded by MS.

The X-ray absorption spectroscopy (XAS) spectra (Ni and Mo K-edge) were collected at the 4B9A beamline in the Beijing Synchrotron Radiation Facility (BSRF). The storage rings of BSRF were operated at 2.5 GeV with a stable current of 400 mA. Using a Si(111) double-crystal monochromator, the data collection was carried out in transmission mode. All spectra were collected in ambient conditions, and the energy of all samples was calibrated by the absorption edge of the Ni foil and Mo foil standard samples. XAS data were analyzed using the Demeter suite of programs[46]. The acquired extended X-ray absorption fine structure (EXAFS) data were processed and normalized according to the standard procedures using the ATHENA program. FEFF fitting was performed using the software Artemis. Subsequently, χ(k) data in the K-space were Fourier transformed to real (R) space using a Hama Fortan code to separate the EXAFS contributions from different coordination shells[47].

The chemical states of the Ni/MTAC samples surface were performed using Thermo-Scientific ESCALAB 250 X-ray photoelectron spectroscopy (XPS) with Al $K_α$ (hν = 1486.6 eV) radiation as the excitation source. The sample after treatment was transferred into sample rod in glove box with $N_2$ atmosphere, and the charging effects were eliminated by correcting the observed spectra with C 1$s$ binding energy value of 284.8 eV.

Raman spectra were recorded on a Renishaw in Via-Reflexm Raman microscope at least three times. A sample of approximately 50 mg were excited using an Ar-ion laser operating at 532 nm.

Scanning electron microscopy (SEM) images were collected using a Hitachi SU8010 high-resolution field emission scanning electron microscope. Transmission electron microscopy (TEM) images were conducted on a Titan G2 60-300 equipment. High-angle annular dark-field scanning transmission electron microscopy (HAADF-STEM) and electron energy loss spectroscopy (EELS) was performed using a JEOL JEM-ARM200F instrument.

In situ diffuse reflectance infrared Fourier transform spectra (In situ DRIFTS) were recorded on a Nicolet iS50 spectrometer equipped with a Harrick Scientific diffuse reflection accessory, a mercury-cadmium-telluride (MCT) detector, and a Hiden HPR-20 R&D mass spectrometer (MS). Prior to the temperature-programmed ethanol reforming (TPER) tests, the as-prepared catalyst was initially preheated from room temperature to 300 °C (10 °C min$^{-1}$) in Ar (40 mL min$^{-1}$) for 0.5 h to remove physiosorbed water and impurities, and then cooled to 50 °C. Once the reference baseline stabilized, the background spectrum was collected. After that, the aqueous ethanol ($H_2O$/$C_2H_5OH$ = 3) was bubbled under the flow of Ar (40 mL min$^{-1}$), meanwhile, the temperature was raised from 50 °C to 600 °C at 25 °C min$^{-1}$. Spectra were collected once per minute. All the possibly generated products, including $H_2$ (m/z = 2), $CH_4$ (m/z = 16), CO (m/z = 28), $CH_3CHO$ (m/z = 29), $CH_3COCH_3$ (m/z = 43), $CO_2$ (m/z = 44), and $C_2H_5OH$ (m/z = 46), were recorded by MS.

## Catalytic testing

Experimental tests were carried out in a vertical fixed bed reactor (8 mm I.D.) under atmospheric pressure. Prior to each reaction, all the samples were reduced at 600 °C for 1 h under 20 vol% $H_2$ in $N_2$ (50 mL min$^{-1}$). Then, 0.30 ± 0.005 g reduced sample was installed into the reactor tube, supported by quartz wool. Before the experiment, the reactor was purged with 40 mL min$^{-1}$ $N_2$ to ensure the removal of oxygen. Subsquently the reactor was heated to a target temperature (450 °C, 500 °C, 550 °C, or 600 °C) with a heating rate of 20 °C min$^{-1}$ under 40 mL min$^{-1}$ $N_2$. Meanwhile, aqueous ethanol (S/E = 3) was injected at 0.005 mL min$^{-1}$ using a HPLC pump (EPP 010 Elite). The feed was heated and evaporated to 120 °C via the heating band. Once the reactor temperature reached the target temperature and stabilized, the vaporized feedstock was flowed into the reactor, and the experiments were kept for 0.5 h. For long-term stability measurements, a SRE duration of 120 h was performed under the same conditions. The concentrations of the outlet gases ($N_2$, $H_2$, CO, $CO_2$, $CH_4$, $C_2H_4$, and $C_2H_6$) were detected by an INFICON Micro GC Fusion equipped with a thermal conductivity detector.

The total production rate ($F_{total}$, mmol/min) of the reformate was calculated according to Eq. 1:

$$F_{total} = F(N_2)/C(N_2)_{out} \tag{1}$$

where $F(N_2)$ denotes the $N_2$ flow rate and $C(N_2)_{out}$ is the nitrogen concentration of the outlet products. The flow rate of the outlet product $i$ ($F(i)_{out}$, mmol/min) and the normalized concentration of the outlet reformate $i$ ($NC(i)_{out}$, vol%) were calculated respectively according to the following Eqs. 2 and 3:

$$F(i)_{out} = F_{total}/C(i)_{out} \tag{2}$$

$$NC(i)_{out} = \frac{C(i)_{out}}{\sum_i C(i)_{out}} \tag{3}$$

where $C(i)_{out}$ represents the concentration of the outlet products $i$ ($i = H_2$, $CO_2$, CO, $CH_4$, $C_2H_4$ or $C_2H_6$). The hydrogen utilization efficiency (HUE, %)[48] was used to comprehensively evaluate the ethanol conversion performance and the $H_2$ selectivity of the entire SRE process and calculated by Eq. 4:

$$HUE = \frac{2 \times F(H_2)_{out}}{6 \times F(C_2H_5OH)_{in} + 2 \times F(H_2O)_{in}} \times 100\% \tag{4}$$

where $F(C_2H_5OH)_{in}$ and $F(H_2O)_{in}$ are the feeding rates of ethanol and steam, respectively.

Thermogravimetry analysis (TGA) was tested to investigate the carbon formation characteristics using a NETZSCH STA 2500 Regulus analyzer. The reacted samples (~40 mg) were placed in a quartz crucible and heated (10 °C min$^{-1}$) from 50 °C to 1000 °C by flowing 100 mL min$^{-1}$air.

## DFT Computation Methods

Density functional theory (DFT) calculations were performed on the Ni-support system using the Vienna Ab initio Simulation package (VASP), employing the projected augmented wave (PAW) method[49]. All periodic boundary calculations were performed with the generalized gradient approximation (GGA) refined by Perdew, Burke, and Ernzerhof (PBE). The vdWs interaction was included by using an empirical DFT-D3 method. The $Ni_{SA}$, $Ni_{NC}$, $Ni_{SA-NC}$ $Ni_{SA-SA}$, and $Ni_{NC-NC}$ represent the models with 1 Ni atom, 12 Ni atoms, 1 Ni atom adjacent to 12 Ni atoms, 1 Ni atom adjacent to 1 Ni atom, and 12 Ni atoms adjacent to 12 Ni atoms respectively. These models were used to investigate the ethanol dehydrogenation. The atoms in the upper two layers of the

$Mo_2TiAlC_2$ surface are allowed to move freely, while the bottom four layers are fixed to simulate the surface of structure. The Monkhorst-Pack-grid-mesh-based Brillouin zone k-points are set as 2×2×1 for all structures with the cutoff energy of 400 eV. The convergence criteria are set as 0.05 eV A$^{-1}$ and 10$^{-4}$ eV in force and energy, respectively.

The free energy calculation of species adsorption (ΔG) is based on Eq. 5:

$$\Delta G = \Delta E + \Delta E_{ZPE} + \Delta H_{0 \rightarrow T} - T \Delta S \tag{5}$$

Herein, $\Delta E$, $\Delta E_{ZPE}$, and $\Delta S$, respectively represent the changes of electronic energy, zero-point energy, and entropy of the intermediate. The $\Delta H_{0 \rightarrow T}$ refers to the change in enthalpy when heating from 0 K to $T$ K (298 K in this work). The entropy of H$^+$+e$^-$ pair is approximately regarded as half of $H_2$ entropy in standard condition.

The energy barrier ($E_a$) is obtained from the electronic energy difference between the transition state ($E_{TS}$) and its corresponding initial state ($E_{IS}$), which is calculated by Eq. 6:

$$E_a = E_{TS} - E_{IS} \tag{6}$$

## Data availability

Other data are available from the corresponding author upon request. All original data needed to evaluate the conclusions in the paper have already been present in the manuscript and the Supplementary Information (including Supplementary Figs. 1–49 and Supplementary Tables 1–6).

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

## Acknowledgements

This work is supported by the National Natural Science Foundation of China (52436006, 52476144), and the Science and Technology Innovation Program of Hunan Province (2022RC1126).

## Author contributions

Z.S., Z.Q.S., and G.J.H. supervised this research and conceived the idea. Z. S. and W.S. performed the catalyst preparation, catalytic experiments, XAS test, DFT calculations, and analyzed the data. W.S. and Z.S. conducted the SEM, HR-TEM, and HAADF-STEM characterizations and analyzed the data with the help of L.R.S., N.F.D., and H.F.Q. Besides, W.S., Z.S., L.R.S., N.F.D., H.F.Q., Z.Q.S., and G.J.H. wrote and revised the manuscript. All authors discussed the results and commented on the paper.

## Competing interests

The authors declare no competing interests.
