## [Transparent Peer Review File · Nature Communications]

Concerted catalysis of single atom and nanocluster enhances bio-ethanol activation and dehydrogenation

Corresponding Author: Professor Graham Hutchings

Version 0:

Reviewer comments:

Reviewer #1

(Remarks to the Author)

Shi et al. have constructed three types of Ni supported catalysts for boosting bio-ethanol reforming, which aimed to demonstrate the synergistic effect of single atoms and clusters. However, the analysis of the electronic relationship between Ni and Mo is insufficient and ambiguous, and the XAS data analysis and DFT calculation of samples is not rigorous. I think the work requires further research and cannot be published in the present form.

1. I think the introduction does not reflect the value of the reaction system studied and the scientific problem solved by the catalyst developed in this work.
2. It is recommended to photograph the sample with a relatively low-resolution TEM to determine the distribution of clusters over a larger area. According to EDS images of 1Ni/MTAC and Fig. 1g, the Ni clusters on the sample are much larger than 0.5nm, or there are some particles having a size of ~ 1 nm. How the authors measure the size and conduct the statistics as shown in Fig. S4?
3. (Line154-156) According to the results of XPS, both Ni⁰ and Ni²⁺ were present in the sample, what about their ratio? How were they formed?
4. XPS results showed that the binding energy of Ni gradually decreased with the increase of Ni AC content (Fig. 2a). However, XANES showed that the binding energy of 1.5Ni/MTAC samples was the highest (Fig. 2c). Why the results were contradictory. In addition, the analysis of Ni valence states by the two characterizations is also inconsistent.
5. The authors analyzed the EXAFS of the samples and concluded that Ni-Mo bonds existed in 0.5Ni/MTAC and 1Ni/MTAC, but Ni-Ni coordination was used when fitting the coordination number (Table S3). Therefore, I doubt about the Ni K-edge fitting process and results. The presence of Ni-Ni bonds in samples containing clusters is understandable, but the authors should distinguish between the coordination of Ni-Ni and Ni-Mo.
6. The result of K-space and R-space fitting should be supplemented in SI. In addition, the Mo K-edge XAFS fitted parameters is missing in the table.
7. In reaction evaluation, different catalysts not only differ in the form of Ni, but also in the amount of active site of the reaction. How did the authors consider this? In addition, the role of Ni single atom and Ni cluster in the reaction needs to be distinguished.
8. In-situ infrared was used to investigate the adsorption and desorption behavior of the catalyst on the substrate/intermediate. The trend of the characteristic peaks of these catalysts is the same at different temperatures, but the difference of adsorption and desorption behavior on the catalysts with different proportion of single atoms and clusters is not distinguished. I think these results indicate that the current characterization results cannot reflect the synergistic effect of single atoms and clusters.
9. Does the naming of the catalyst in the DFT calculation model match the catalyst described earlier? There was ambiguity.
10. The DFT calculation work is rough. According to the analysis of the surface reaction path of Ni¹/MTAC(004) and Ni¹²/MTAC(004) models, both Ni single atoms and clusters can serve as adsorption sites for the reactants. Therefore, more computational data are needed to support the existing reaction path and optimization process for the Ni¹/Ni¹²/MTAC(004), which includes both single atoms and clusters.

Reviewer #2

(Remarks to the Author)

This paper presents interesting and useful results devoted to elucidation of the role of Ni dispersion in xNi/Mo₂TiAlC₂

catalysts on their performance in ethanol steam reforming. A positive effect of combination of isolated Ni atoms and nanoclusters was reliably demonstrated. A combination of modern research techniques including structural and spectral methods as well as DFT calculations was applied providing solid foundations for solving the article tasks. Presentation of results is good, their analysis is proper, description good, conclusions are sound. Hence, this paper can be accepted for publication. At proofs reading stage I would recommend to add description of samples preparation procedure for XPS studies to be sure of any possible impact of samples contact with air and impurities on the spectra.

Reviewer #3

(Remarks to the Author)

The main basis of this work is the preparation of nickel systems in different chemical states, namely single atoms and nanoclusters of Ni. To demonstrate this, the authors have relied mainly on the results obtained by TEM, XPS and XAS characterization techniques. In my opinion, these results do not at all prove the presence of these species.

Regarding the HAADF-STEM images obtained (Figure 1), the presence of single atoms, always difficult to detect, requires in this case a much more precise analysis to be able to ensure the existence of these kind of species.

Regarding the results obtained by XPS, it is surprising that the presence of oxidized nickel species is not discussed in the framework of the supposed existence of the different types of particles, single atoms and nanoclusters.

But the conclusions of the study by XAS are especially surprising. First of all, the differences observed in the XANES spectra (Figure 2C) must be justified considering the resolution of the recorded spectrum, although this value is not mentioned in the manuscript.

And secondly, the results obtained by EXAFS are not credible at all. The mere visual inspection of the different TF lines (Figure 2E) does not correspond to the CN values obtained by mathematical fitting. The CN value = 0.32 obtained for the 0.5Ni/MTAC sample is completely absurd in view of the intensity of the peak at 2.09 corresponding to Ni-Ni neighbors (Figure 2e, green line). But what is more, even in the TF of this sample the presence of successive spheres of coordination can be clearly intuited by the presence of peaks in the interval between 3 and 5 Å (compare for example the green and red lines corresponding to the samples 0.5Ni and 1Ni, respectively). All of this is incompatible with the presence of single nickel atoms. Therefore, in my opinion, since the existence of SA Ni particles in the 0.5Ni catalyst is doubtful, the general conclusions of the work are not at all justified.

Version 1:

Reviewer comments:

Reviewer #1

(Remarks to the Author)

I'm satisfied with the authors' response, and the current version can be accepted.

Reviewer #2

(Remarks to the Author)

Required corrections were made, so paper now can be accepted. Since it is revised version, it is not essential to repeat that all is proper, interesting for readers and significant

Reviewer #3

(Remarks to the Author)

Thank you for the answers to my remarks. In my opinion, the authors partially resolve the issues, but serious doubts remain about the main conclusions of the article.

Regarding the XPS results, the authors now mention and discuss the existence of Ni²⁺ species, and they directly assign it to Ni single atoms. But this assignment must be further demonstrated. Also, they claim that the observed shift of the signal of reduced Ni species, assign to nanoclusters, is due to the transition from isolated single atoms to nanoclusters. I cannot see how this transition can modify the BE of this signal.

Regarding the results obtained through XAS, I do not understand the meaning of the phrase "while that of XAS results show different regularity" (page 30). What was wrong with the first measurements?

XANES included in Figure 2a show similar trends for the 3 samples, indicating no important differences between the structure of Ni particles. Thus, these result do not allow to conclude about the nature of the supposed single atoms. The TFs in Figure 2e point in the same direction. They do not indicate or rule out the existence of single atoms. Finally, the discrepancies between the experimental signals and the fitting included in Figure S14 seem to be similar or even higher to the Ni-Mo contribution. Therefore, it is difficult to draw any conclusions about this kind of interaction.

A point-to-point response to the reviewer's comments

Reviewer #1:

Shi et al. have constructed three types of Ni supported catalysts for boosting bio-ethanol reforming, which aimed to demonstrate the synergistic effect of single atoms and clusters. However, the analysis of the electronic relationship between Ni and Mo is insufficient and ambiguous, and the XAS data analysis and DFT calculation of samples is not rigorous. I think the work requires further research and cannot be published in the present form.

Our response: Thank you very much for your valuable feedback and constructive comments. We appreciate your thoughtful assessment of our manuscript and the opportunity to improve its clarity and scientific rigor. Based on your insightful comments and valuable advice, we have intensively revised the entire manuscript. This includes the clarification of the electronic relationship between Ni and Mo, in-depth analysis of XAS measurements, and further calculation of DFT by performing new configurations of single-atom-with-single-atom and nanocluster-with-nanocluster scenarios. The detailed revisions are provided in the attached file titled "Revised Manuscript." Below, we present our detailed responses to your specific questions and suggestions as well as our detailed revisions.

Specific response:

Comment 1: I think the introduction does not reflect the value of the reaction system studied and the scientific problem solved by the catalyst developed in this work.

Our response: Thank you very much for your valuable feedback. Based on your comment, we have carefully updated the Introduction part. We suppose that the value of the reaction system studied and the scientific problem solved by this work are demonstrated in the following two points:

i) Traditional ethanol reforming studies typically employ high-Ni-loading catalytic systems, which result in low Ni atom utilization as well as catalyst sintering and agglomeration. To the best of our knowledge, very few studies have explored ethanol

reforming using Ni-based atomically dispersed or single-atom catalyst, while this study demonstrates the feasibility of ethanol reforming with a low-Ni-loading catalytic system;

ii) Previous methods typically employ sole single atoms or nanoclusters, while this study provides new insights into the construction of dual active sites (single atoms with nanoclusters) for boosted hydrogen production from ethanol, offering preliminary evidence of the synergistic effect between single atoms and nanoclusters. It is revealed that single atoms facilitate ethanol adsorption and activation, while nanoclusters lower the dehydrogenation energy barriers of intermediates, thereby collectively promoting ethanol reforming to hydrogen.

Detailed revisions: In response to your comments, we have thoroughly revised the Introduction section to better highlight the value of the reaction system and the scientific problem solved by the catalyst developed in our work. Detailed revisions of Introduction are provided below.

Line 48: Nevertheless, to the best of our knowledge, very few studies have explored ethanol reforming using a low-Ni-loading catalytic system. Reducing Ni loading while maximizing atom-utilization efficiency, even enhancing the catalytic performance, remains a significant yet challenging task.

Line 60: However, while SACs benefit from highly dispersed and isolated active sites, the limited overall number compared to nanoparticles or nanoclusters can constrain their catalytic activity. This trade-off between atomic dispersion and site density poses a challenge for SACs in reactions requiring abundant active sites. In the case of ethanol reforming to hydrogen, the applicability and feasibility of Ni-based SACs remain unclear. Specifically, their effects on ethanol conversion and hydrogen selectivity require further exploration.

Line 69: Atomic clusters are energetically favored and stable than single atoms under working conditions due to their lower surface free energy and the formation of strong metal-metal bonds. Previous studies have primarily focused on constructing either single-atom sites or cluster sites. Coupling nanoclusters (NCs) with single atoms (SAs) offers great potential for tailoring the electronic properties and coordination

environments of the combined catalyst, thereby promoting catalytic activity or selectivity through a synergistic mechanism.

Comment 2: It is recommended to photograph the sample with a relatively low-resolution TEM to determine the distribution of clusters over a larger area. According to EDS images of 1Ni/MTAC and Fig. 1g, the Ni clusters on the sample are much larger than 0.5 nm, or there are some particles having a size of ~1 nm. How the authors measure the size and conduct the statistics as shown in Fig. S4?

Our response: Thank you very much for your valuable advice. Based on your comment, we have photographed more spots with both low-resolution and high-resolution TEM to further determine the distribution of the Ni single atoms and nanoclusters. Specifically, we measured and analyzed the overall distribution of clusters using low-magnification TEM images, followed by high-magnification TEM imaging to accurately count single atoms. For the 0.5Ni/MTAC sample, a total of 135 sites were analyzed, among which 120 were single atoms; For the 1Ni/MTAC sample, a total of 155 sites were analyzed, among which 59 were single atoms; For the 1.5Ni/MTAC sample, a total of 114 sites were analyzed, among which 17 were single atoms. The above results correspond to the single atom frequencies of 89%, 38%, and 15%, respectively.

The size of 0.5Ni/MTAC, 1Ni/MTAC, and 1.5Ni/MTAC samples were measured and counted by the software DigitalMicrograph, and the updated results were shown below. We agree that single atoms and nanoclusters exist together for all the samples, while single atoms and nanoclusters are the main forms of 0.5Ni/MTAC and 1.5Ni/MTAC samples, respectively.

Detailed revisions: In response to your comments, we have collected both low-resolution and high-resolution-TEM images, determined the distribution of clusters over a larger area, and calculated the particle size and active site distributions of 0.5Ni/MTAC, 1Ni/MTAC, and 1.5Ni/MTAC. The detailed results are provided below as well as put into the supplementary materials.

Supplementary Figure 3 a, b relatively low-resolution TEM, c-d high-resolution-TEM images of 1Ni/MTAC.

Supplementary Figure 5. Size distribution of 1Ni/MTAC catalyst.

Supplementary Figure 4. Active site distributions of 0.5 Ni/MTAC, 1 Ni/MTAC, and 1.5Ni/MTAC.

Comment 3: (Line154-156) According to the results of XPS, both Ni^0 and Ni^{2+} were present in the sample, what about their ratio? How were they formed?

Our response: Thank you very much for your comment. We assigned the XPS peaks into two peaks, corresponding to $\text{Ni}^{\delta+}$ ($0 < \delta < 2$) and Ni^{2+} . Based on your comment, we calculated the ratios of $\text{Ni}^{\delta+}$ and Ni^{2+} of samples with different Ni loadings, and provided further analysis on the XPS results. From Table 3 below, we can see that the ratios of $\text{Ni}^{\delta+}$ -to- Ni^{2+} increase with Ni loadings. Combined with the hydrogen production performance, we believe that an appropriate proportion of $\text{Ni}^{\delta+}$ -to- Ni^{2+} is a key factor for the efficient catalysis of ethanol to hydrogen.

Supplementary Table 3. Relative proportion of the Ni species for the as-prepared catalysts.

Samples	$\text{Ni}^{\delta+}/\text{Ni}^{\delta+}+\text{Ni}^{2+}$ (%) ^a	$\text{Ni}^{2+}/\text{Ni}^{\delta+}+\text{Ni}^{2+}$ (%) ^a	$\text{Ni}^{\delta+}/\text{Ni}^{2+}$
0.5Ni/MTAC	18.5	81.5	0.227
0.75Ni/MTAC	33.9	66.1	0.513
1Ni/MTAC	36.8	63.2	0.582
1.25Ni/MTAC	38.7	61.3	0.631
1.5Ni/MTAC	39.2	60.8	0.645

^a Determined by sub-peak areas of XPS spectra.

Detailed revisions: We give a further analysis on the XPS results, as provided below:

X-ray photoelectron spectroscopy (XPS) analysis of $x\text{Ni}/\text{MTAC}$ revealed the coexistence of $\text{Ni}^{\delta+}$ (852.4→853.0 eV, $0 < \delta < 2$) and Ni^{2+} (855.4→855.8 eV) species. The $\text{Ni}^{\delta+}$ species correspond to Ni nanoclusters, while the Ni^{2+} species, exhibiting a higher oxidation stage, are attributed to Ni single atoms due to the electronic structure modifications induced by their atomic dispersion (Fig. 2a and Supplementary Fig. 5)^{1, 2}.

As the Ni content increases from 0.5 to 1.5 wt%, the proportion of $\text{Ni}^{\delta+}$ species rises, accompanied by a gradual shift of its peaks to lower binding energies (Supplementary Table 3). This shift suggests a weakening of electron transfer within Ni, consistent with the transition from isolated single atoms to nanoclusters.

Fig. 2a High-resolution Ni 2p spectra of $x\text{Ni}/\text{MTAC}$ ($x=0.5, 0.75, 1, 1.25, \text{ and } 1.5$).

References

- [1] Haoquan Guo, Jiwu Zhao, Yu Chen, Xinyu Lu, Yue Yang, Chenrong Ding, Lizhi Wu, Li Tan, Jinlin Long, Guohui Yang, Yu Tang, Noritatsu Tsubaki, Xiaoli Gu, Mechanistic insights into hydrodeoxygenation of lignin derivatives over Ni Single atoms supported on Mo_2C , *ACS Catalysis*, 2024, 14 (2): 703-717.

[2] Xin Zhang, Wen-Xiao Liu, Yi-Wen Zhou, Ze-Da Meng, Li Luo, Shou-Qing Liu, Single-atom nickel anchored on surface of molybdenum disulfide for efficient hydrogen evolution, *Journal of Electroanalytical Chemistry*, 2021, 894: 115359.

Comment 4: XPS results showed that the binding energy of Ni gradually decreased with the increase of Ni AC content (Fig. 2a). However, XANES showed that the binding energy of 1.5Ni/MTAC samples was the highest (Fig. 2c). Why the results were contradictory. In addition, the analysis of Ni valence states by the two characterizations is also inconsistent.

Our response: Thank you for your insightful comment. We agree that there is an apparent contradiction between the trends observed in XPS and XANES regarding the relationship between Ni valence states and loading content. We supposed that this discrepancy may be attributed to instrumental errors during the measurements of XPS or XAS. To further verify the results, we repeated both XPS and XAS measurements. The repeated XPS results are in agreement with the previous ones, while that of XAS results show different regularity, confirming the reliability of XPS. For the second-time XAS results this time, the absorption edges of three samples were all located within the Ni foil and NiO, i.e., the repeated XAS results now align with the XPS findings, implying the positive charge of Ni. Therefore, the previous inconsistency has been resolved.

Detailed revisions: We sincerely appreciate your insightful comments, and based on your comments, we have intensively revised the previous descriptions and updated the related figures, as shown below:

The electronic structure and coordination environment of Ni and Mo species was further determined by X-ray absorption spectroscopy (XAS) measurements. The normalized X-ray near-edge absorption spectra (XANES) of Ni K-edge are presented in Fig. 2c. The absorption edges of three samples were all located between Ni foil and NiO, suggesting the positive charge of the Ni atoms, which is in agreement with the XPS results³⁴. Besides, the adsorption edge of 1.5Ni/MTAC shows the smallest shift to higher energy of the three catalysts, indicative of the lowest electron density of

interfacial Ni atom. As shown in normalized XANES spectra of Mo K-edge (Fig. 2d), the absorption edge of all samples is located between the Mo foil and MoO₂, suggesting the positively charged state of Mo, which is in good agreement with the XPS results.

The coordination environments were further analyzed by the extended X-ray absorption fine structure (EXAFS) at the K-edge of Ni and Mo. The Ni K-edge EXAFS of Ni foil shows a main peak at 2.09 Å (not phase-corrected), which was attributed to the Ni-Ni scattering path (Fig. 2e and Supplementary Figs. 12, 13). With the decrease in the Ni loadings, the peak gradually shifted to longer R value, owing to the formation of Ni-Mo path. Compared to Ni foil, the intensity of the Ni-Ni scattering path gradually diminished, confirming the interaction between Ni and MTAC support with the formation of Ni-Mo path, corresponding to coordination number reduction from 12.0 to 9.6, 8.5, and 6.7 (Supplementary Table 4). The lengths of Ni-Ni/Mo bonds in 0.5, 1, and 1.5Ni/MTAC further increased compared to Ni foil, testifying the formation of Ni-Mo bonds. Moreover, with the increase of Ni content, the coordination number of Ni-Mo decreases, resulting in the weakening of metal-support interaction.

For Mo-edge EXAFS, all the Ni/MTAC samples exhibited strong Mo-C, Mo-Ni, and Mo-Mo scattering (Fig. 2f and Supplementary Figs. 14, 15)³⁵. The 2.27 Å contribution (not phase-corrected) was attributed to the Mo-Ni scattering, which is slightly shifted compared with the Mo-Mo scattering in Mo foil (Supplementary Table 5)³⁶. To precisely clarify the atomic dispersion and coordination conditions of Ni and Mo, the wavelet transform of EXAFS spectra were analysed (Supplementary Figs. 16, 17), further confirming the formation of the Ni-Mo coordination.

Fig. 2 Electronic state and atomic structure characterization. **a** High-resolution Ni 2p spectra and **b** Mo 3d spectra. **c** The normalized Ni K-edge XANES spectra. **d** The normalized Mo K-edge XANES spectra. **e** Corresponding Fourier-transformed R-space spectra (solid lines) and fits (open points) of Fig. 2c. **f** Corresponding Fourier-transformed R-space spectra of Fig. 2d.

Comment 5: The authors analyzed the EXAFS of the samples and concluded that Ni-Mo bonds existed in 0.5Ni/MTAC and 1Ni/MTAC, but Ni-Ni coordination was used when fitting the coordination number (Table S3). Therefore, I doubt about the Ni K-edge fitting process and results. The presence of Ni-Ni bonds in samples containing clusters is understandable, but the authors should distinguish between the coordination of Ni-Ni and Ni-Mo.

Our response: Thank you very much for your valuable comment. Based on your comment, we updated the fitting of Ni and Mo K-edge EXAFS spectra by using both Ni-Ni and Ni-Mo paths.

Detailed revisions: The detailed revisions are provided below, which includes the fitting results of Ni k-edge Fourier-transform EXAFS spectra on K-space and R-space, wavelet transformed EXAFS spectra, as well as Ni K-edge EXAFS fitted parameters for various catalysts, as shown below:

Supplementary Figure 12. XAS analysis of as-prepared samples. a Fourier transform of Ni K-edge EXAFS spectra in K-space. The fitting results of Ni K-edge Fourier-transform EXAFS spectra on K-space of **b** 0.5Ni/MTAC, **c** 1Ni/MTAC, and **d** 1.5Ni/MTAC.

Supplementary Figure 13. The fitting results of Ni k-edge Fourier-transform EXAFS spectra on R-space of **a** 0.5Ni/MTAC, **b** 1Ni/MTAC, and **c** 1.5Ni/MTAC.

Supplementary Figure 16. Wavelet transformed EXAFS spectra. Ni K-edge EXAFS spectra for **a** Ni foil, **b** 0.5Ni/MTAC, **c** 1Ni/MTAC, and **d** 1.5Ni/MTAC.

Supplementary Table 4. Ni K-edge EXAFS fitted parameters for as-prepared catalysts.

Samples	shell	CN ^a	R(Å) ^b	$\sigma^2 \times 10^2$ (Å ²) ^c	ΔE_0 (eV) ^d	R factor ^e
Ni foil	Ni-Ni	12.0*	2.482	0.61	7.6	0.0008
NiO	Ni-O	5.9	2.078	0.51	-3.3	0.0055
	Ni-Ni	12.2	2.949	0.59		
0.5Ni/MTAC	Ni-Ni	6.7	2.477	0.82	2.3	0.0051
	Ni-Mo	2.3	2.720			
1Ni/MTAC	Ni-Ni	8.5	2.489	0.72	6.0	0.0090
	Ni-Mo	1.3	2.790			
1.5Ni/MTAC	Ni-Ni	9.6	2.490	0.81	4.1	0.0052
	Ni-Mo	0.4	2.790			

^a Coordination numbers.

^b Bond distance.

^c Debye-Waller disorder factor

^d adsorption edge offset.

^e goodness of fit.

* This value was fixed during EXAFS fitting, based on the known structure of Ni.

Comment 6: The result of K-space and R-space fitting should be supplemented in SI. In addition, the Mo K-edge XAFS fitted parameters is missing in the table.

Our response: Thank you very much for your valuable advice. Based on your suggestion, we supplemented the fitting result of K-space and R-space. Moreover, the Mo K-edge XAS fitted parameters is added.

Detailed revisions: Please see the detailed revisions below.

Supplementary Figure 14. XAS analysis of as-prepared samples. **a** Fourier transform of Mo K-edge EXAFS spectra in K-space. The fitting results of Mo k-edge Fourier-transform EXAFS spectra on K-space of **b** 0.5Ni/MTAC, **c** 1Ni/MTAC, and **d** 1.5Ni/MTAC.

Supplementary Figure 15. The fitting results of Mo k-edge Fourier-transform EXAFS spectra on R-space of **a** 0.5Ni/MTAC, **b** 1Ni/MTAC, and **c** 1.5Ni/MTAC.

Supplementary Figure 17. Wavelet transformed EXAFS spectra. Mo K-edge EXAFS spectra for **a** Mo₂C. **b** 0.5Ni/MTAC. **c** 1Ni/MTAC, and **d** 1.5Ni/MTAC.

Supplementary Table 5. Mo K-edge EXAFS fitted parameters for as-prepared 0.5, 1, and 1.5Ni/MTAC catalysts.

Samples	shell	CN ^a	R(Å) ^b	$\sigma^2 \times 10^2$ (Å ²) ^c	ΔE_0 (eV) ^d	R factor ^e
Mo foil	Mo-Mo	8.0*	2.717	0.38	5.3	0.0013
	Mo-Mo	6.0*	3.130	0.34		
Mo ₂ C	Mo-C	2.7	2.105	0.40	-7.6	0.0148
	Mo-Mo	6.4	2.982	0.75		
0.5Ni/MTAC	Mo-C	3.0	2.083	0.38	-14.5	0.0145
	Mo-Ni	4.4	2.788	0.71		
	Mo-Mo	6.5	3.237	0.72		
1Ni/MTAC	Mo-C	3.0	2.085	0.43	-13.4	0.0175
	Mo-Ni	4.3	2.788	0.71		
	Mo-Mo	6.3	3.237	0.71		
1.5Ni/MTAC	Mo-C	2.9	2.073	0.34	-12.9	0.0146
	Mo-Ni	4.0	2.789	0.68		
	Mo-Mo	5.8	3.237	0.68		

^a Coordination numbers.

^b Bond distance.

^c Debye-Waller disorder factor

^d adsorption edge offset.

^e goodness of fit.

* This value was fixed during EXAFS fitting, based on the known structure of Mo.

Comment 7: In reaction evaluation, different catalysts not only differ in the form of Ni, but also in the amount of active site of the reaction. How did the authors consider this? In addition, the role of Ni single atom and Ni cluster in the reaction needs to be distinguished.

Our response: Thank you very much for your valuable advice. According to our experimental investigation, catalyst characterization, and DFT calculation, the adjacent Ni single atom enhances bio-ethanol adsorption and activation, while Ni nanoclusters contributes to lowering the energy barriers of intermediate such as CH₃CHO* dehydrogenation.

Detailed revisions: Regarding your suggestion for reaction performance evaluation, we first estimated the amount of the active sites for 0.5Ni/MTAC, 1Ni/MTAC, and 1.5Ni/MTAC samples. It is considered that: i) a same weight of catalyst is added (1.000 g); ii) the proportions of single atoms are in accordance with the statistical data from TEM (89%, 38%, and 15%); iii) There are 50 atoms for a Ni nanoclusters with a size of around 0.6 nm, by taking *Physical Chemistry Chemical Physics*, 2010, 12: 5562-5574 as a reference.

Based on the above assumptions, for 0.5Ni/MTAC sample, the numbers of cluster sites would be 8.81×10^{17} , the numbers of single-atom sites would be 7.13×10^{18} , the numbers of Ni in nanoclusters would be 4.41×10^{19} , and the numbers of Ni in single atoms would be 7.13×10^{18} , total active site number would be **8.01×10^{18}** ; for 1Ni/MTAC sample, the numbers of cluster sites would be 1.27×10^{18} , the numbers of single-atom sites would be 3.90×10^{19} , the numbers of Ni in nanoclusters would be 6.36×10^{19} , and the numbers of Ni in single atoms would be 3.90×10^{19} , total active site number would be **40.27×10^{18}** ; for 1.5Ni/MTAC sample, the numbers of cluster sites would be 2.62×10^{18} , the numbers of single-atom sites would be 2.31×10^{19} , the numbers of Ni in

nanoclusters would be 1.31×10^{20} , and the numbers of Ni in single atoms would be 2.31×10^{19} , total active site number would be 25.72×10^{18} . Therefore, we suppose that the calculation of the total active site number can to some extent explain the better performance of 1Ni/MTAC sample.

Supplementary Figure 22 Estimated total active site numbers of 0.5Ni/MTAC, 1Ni/MTAC, and 1.5Ni/MTAC catalysts.

Besides, we also evaluated the reaction performance using hydrogen production efficiency per unit Ni atom. This method normalizes the added Ni content to evaluate the hydrogen production rate per unit Ni molar amount. The related results were provided below, which sufficiently demonstrates the superiority of the 1Ni/MTAC catalyst and the synergistic effect between the Ni single atoms and nanoclusters.

Fig. 3h. Comparison of unit hydrogen yield of all the as-prepared catalysts.

Supplementary Figure 30. Unit reaction rate of 1Ni/MTAC and 10Ni/MTAC. Activity test conditions: 1 atm, 550 °C, S/E=3, liquid feeding rate 0.005 mL min⁻¹, N₂ flow rate 40 mL min⁻¹.

Comment 8: In situ infrared was used to investigate the adsorption and desorption behavior of the catalyst on the substrate/intermediate. The trend of the characteristic peaks of these catalysts is the same at different temperatures, but the difference of adsorption and desorption behavior on the catalysts with different proportion of single atoms and clusters is not distinguished. I think these results indicate that the current characterization results cannot reflect the synergistic effect of single atoms and clusters.

Our response: Thank you for your valuable comments. Upon further analysis of the previous results, we observed that the adsorption peaks of intermediate species were either very small or showed inverted peaks, indicating relatively weak adsorption of ethanol on all the catalysts. We believe this could be solved by changing the testing method. To enhance the detection signal of ethanol intermediates, we have redesigned the in situ infrared testing procedure. In the previous experiments, ethanol was adsorbed onto the catalysts, followed by purging until stabilization, and then conducted the temperature-programmed surface reaction. The new approach involves continuously feeding ethanol into the system and performing temperature-programmed experiments. This method ensures a higher concentration of intermediates, allowing for better observation of the adsorption and desorption behavior of the catalyst.

Detailed revisions: Based on your comments, we have re-designed the in situ DRIFTS experiments, and we name it temperature-programmed ethanol reforming (TPER) measurements. The detailed procedure are as follows: Prior to the temperature-programmed ethanol reforming (TPER) tests, the as-prepared catalyst was initially preheated from room temperature to 300 °C (10 °C min⁻¹) in Ar (40 mL min⁻¹) for 0.5 h to remove physisorbed water and impurities, and then cooled to 50 °C. Once the reference baseline stabilized, the background spectrum was collected. After that, the aqueous ethanol (H₂O/C₂H₅OH=3) was bubbled under the flow of Ar (40 mL min⁻¹), meanwhile, the temperature was raised from 50 °C to 600 °C at 25 °C min⁻¹. Spectra were collected once per minute. All the possibly generated products, including H₂ (m/z = 2), CH₄ (m/z = 16), CO (m/z = 28), CH₃CHO (m/z = 29), CH₃COCH₃ (m/z = 43), CO₂ (m/z = 44), and C₂H₅OH (m/z = 46), were recorded by MS.

For TPER experiments, important reaction intermediates such as CH₃CH₂O*, CH₃CHO*, CH₃COO* were captured, and the results are provided in Fig. 4e-4g. **For 0.5Ni/MTAC sample**, the relatively low intensities of the CO₂ peaks at 2346 cm⁻¹ and 2310 cm⁻¹ indicate its inferior performance in deep dehydrogenation and reformation with hydrogen and CO₂ production. Additionally, the stretching vibration at 1734 cm⁻¹ is attributed to the presence of the C₂H₄O* species. The prominent peak at this position suggests that for the 0.5Ni sample, which has a higher proportion of single atoms, the energy barrier for the C₂H₄O* dehydrogenation step is relatively high, which is consistent with the DFT calculation results. **For 1.5Ni/MTAC sample**, the peak at 2987 cm⁻¹ corresponds to the stretching vibration of the C₂H₅O* functional group, indicating that the C₂H₅O* dehydrogenation step is the rate-limiting step at low temperatures (<350 °C). As the temperature increases, the CO₂ peak intensity significantly increases, suggesting excellent ethanol reforming performance for hydrogen production. This result aligns with the DFT calculations, where the C₂H₅O* dehydrogenation step in nanoclusters exhibits the highest energy barrier. **For 1Ni/MTAC sample**, when the temperature is above 350 °C, the peak intensities of C₂H₅O* and C₂H₄O* are relatively low, while that of CO₂ is remarkably observed, demonstrating its superiorities in intermediate dehydrogenation and reformation.

Fig. 4e. In situ DRIFTS of 0.5Ni/MTAC.

Fig. 4f. In situ DRIFTS of 1Ni/MTAC.

Fig. 4g. In situ DRIFTS results using 1.5Ni/MTAC catalyst.

Supplementary Figure 41 Comparisons of TPER experimental results at 350 °C.

Comment 9: Does the naming of the catalyst in the DFT calculation model match the catalyst described earlier? There was ambiguity.

Our response: Thank you very much for your valuable advice. Based on your suggestion, we have updated the names of the catalysts in the DFT calculation. Five scenarios were considered: which are i) 1 Ni atom (single atom); ii) 12 Ni atoms (nanocluster); iii) 1 Ni atom adjacent to 12 Ni atoms; iv) 1 Ni atom adjacent to 1 Ni atom; and v) 12 Ni atoms adjacent to 12 Ni atoms, corresponding to Ni_{SA}, Ni_{NC}, Ni_{SA-NC}, Ni_{SA-SA}, and Ni_{NC-NC}, respectively. The above updates have been implemented throughout the entire manuscript. We sincerely appreciate your constructive feedback again.

Detailed revisions: We have revised the manuscript to ensure consistent naming throughout, clarifying the catalyst types described in the main text and figures by replacing Ni₁, Ni₁₂, Ni_{1Ni12} by Ni_{SA}, Ni_{NC}, and Ni_{SA-NC}. This revision addresses the ambiguity and ensures clear alignment between the catalyst descriptions and the DFT calculation model.

Comment 10: The DFT calculation work is rough. According to the analysis of the surface reaction path of Ni₁/MTAC(004) and Ni₁₂/MTAC(004) models, both Ni single

atoms and clusters can serve as adsorption sites for the reactants. Therefore, more computational data are needed to support the existing reaction path and optimization process for the Ni₁/Ni₁₂/MTAC(004), which includes both single atoms and clusters.

Our response: Thank you very much for your valuable advice. Based on your comments, we conducted more DFT calculations by configuring the models with both single atoms and both clusters. Specifically, we calculated the reaction energy barrier for ethanol dehydrogenation over Ni_{SA-SA}/MTAC (both single atoms) and Ni_{NC-NC}/MTAC (both clusters) models.

Detailed revisions: To make a clear comparison, the results of Ni_{SA-SA}/MTAC, Ni_{SA-NC}/MTAC, and Ni_{NC-NC}/MTAC are put together, as shown below:

Fig. 6 DFT calculations of CH₃CH₂OH dehydrogenation. Potential diagram from DFT calculations of CH₃CH₂OH dehydrogenation over Ni_{SA-SA}/MTAC(004) (orange), Ni_{NC-NC}/MTAC(004) (blue), and Ni_{SA-NC}/MTAC(004) (red).

When two Ni single atoms (Ni_{SA-SA}) and two Ni nanoclusters (Ni_{NC-NC}) are introduced, the energy barriers for the transition states remain basically similar

tendencies, i.e., the computational results align with our previous findings. This indicates that the adsorption of the intermediate on the supported Ni single atom is still strong, while the introduction of nanocluster modulates an appropriate adsorption for ethanol activation as well as lower the energy barriers for the dehydrogenation of ethanol and other intermediates. Specifically, the barrier of 1.5Ni/MTAC for $C_2H_5O^*$ hydrogenation to $C_2H_4O^*$ is still high, this is in agreement with the in situ DRIFTS results. Besides, the rate-determining step of all catalyst remains the dehydrogenation of acetaldehyde, with the transition state barrier of 0.5Ni/MTAC and 1.5Ni/MTAC decreasing from 2.97 eV \rightarrow 2.89 eV and 1.64 eV \rightarrow 1.54 eV. Overall, Ni_{SA-NC} models with single atom and cluster co-adsorption is the best, and its potential barrier is the smallest of 1.28 eV. The synergy of single atoms and nanoclusters results in a moderate adsorption of the intermediate species and sharply decreased energy barriers for intermediates dehydrogenation on Ni_{SA-NC}/MTAC, thus boosting the catalytic activity of bio-ethanol reforming.

Reviewer #2:

This paper presents interesting and useful results devoted to elucidation of the role of Ni dispersion in $x\text{Ni}/\text{Mo}_2\text{TiAlC}_2$ catalysts on their performance in ethanol steam reforming. A positive effect of combination of isolated Ni atoms and nanoclusters was reliably demonstrated. A combination of modern research techniques including structural and spectral methods as well as DFT calculations was applied providing solid foundations for solving the article tasks. Presentation of results is good, their analysis is proper, description good, and conclusions are sound. Hence, this paper can be accepted for publication. At proofs reading stage I would recommend to add description of samples preparation procedure for XPS studies to be sure of any possible impact of samples contact with air and impurities on the spectra.

Our response: We appreciate the reviewer for the very positive comments, which allowed us to improve the paper quality after appropriate revisions. Based on your comments and useful advice, we have added the description of samples preparation procedure for XPS studies.

Detailed revisions: Detailed revisions are provided below:

Line 464: “The chemical states of the Ni/MTAC samples surface were performed using Thermo-Scientific ESCALAB 250 X-ray photoelectron spectroscopy (XPS) with Al $K\alpha$ ($h\nu = 1486.6$ eV) radiation as the excitation source. The sample after treatment was transferred into sample rod in glove box with N_2 atmosphere, and the charging effects were eliminated by correcting the observed spectra with C 1s binding energy value of 284.8 eV.”

Reviewer #3:

The main basis of this work is the preparation of nickel systems in different chemical states, namely single atoms and nanoclusters of Ni. To demonstrate this, the authors have relied mainly on the results obtained by TEM, XPS and XAS characterization techniques. In my opinion, these results do not at all prove the presence of these species.

Our response: Thank you very much for your insightful comments and critical evaluation of our work. We sincerely appreciate your time and effort in reviewing our manuscript. Your feedback has been extremely valuable in helping us refine our analysis and further clarify our conclusions.

Regarding your concern about the identification of Ni single atoms and nanoclusters, we understand the importance of rigorous characterization. To further validate the coexistence of these species, we have conducted more characterizations (AC-STEM and in situ DRIFTS) and updated the XAS results to further prove the advantages of the combination of single atoms and nanoclusters. These revisions and updates provide a more comprehensive and convincing demonstration of the structural characteristics of our system.

We are truly grateful for your constructive critique, which has helped us improve the clarity and scientific rigor of our study. We hope that with these revisions, our findings will be more convincingly presented, and we greatly appreciate any further suggestions you may have. Thank you again for your valuable time and expertise. The detailed revisions can be seen in the attached file named “Revised Manuscript”. The point-by-point responses to the questions/suggestions are shown as below.

Specific responses:

Comment 1: Regarding the HAADF-STEM images obtained (Figure 1), the presence of single atoms, always difficult to detect, requires in this case a much more precise analysis to be able to ensure the existence of these kind of species.

Our response: Thank you for your valuable comment. We fully understand that the identification of single atoms in HAADF-STEM images is challenging and requires careful analysis. To address this concern, we have conducted a more detailed

investigation by acquiring additional aberration-corrected HAADF-STEM images with higher precision. These new images, combined with the corresponding STEM-EDS mapping and EELS results, indicate that the brighter regions correspond to the distribution of Ni elements.

In addition, based on the more photographed low-resolution and high-resolution TEM images, we also further updated the distributions of the Ni single atoms and nanoclusters under. For the 0.5Ni/MTAC sample, a total of 135 sites were analyzed, among which 120 were single atoms; For the 1Ni/MTAC sample, a total of 155 sites were analyzed, among which 59 were single atoms; For the 1.5Ni/MTAC sample, a total of 114 sites were analyzed, among which 17 were single atoms.

Detailed revisions: The updated results are provided below.

Supplementary Figure 3 a, b TEM. c-d high-resolution-TEM images of 1Ni/MTAC.

Supplementary Figure 5. Active site distributions of 0.5 Ni/MTAC, 1 Ni/MTAC, and 1.5Ni/MTAC.

Supplementary Figure 6. EDS mapping results of 1Ni/MTAC sample for Ni, Mo, Ti, and Al elements.

For further identify the Ni species, we performed EELS analysis in the bright regions where the clusters are located. The characteristic Ni signal observed in these areas provides additional evidence, corroborating that the bright spots distributed are Ni. Additionally, the position of L3 and L2 illustrate the existence of high-valence Ni, which is in agreement with the XPS results. The same method has also been reported in other literature, as shown below (Daems et al, *Sustainable Energy & Fuels*, 2020, 4: 1296), where, when the EELS signal of a single atom is not strong enough, EELS analysis of clusters or particles with similar brightness is performed to determine the

elemental composition of single atoms with the same brightness. Therefore, combined with AC-STEM, EDS mapping, and EELS results, we concluded that the detected single atoms on the bright area is Ni.

[Figure Redacted]

Supplementary Figure 3. EELS analysis on nanoclusters in a bright area of 1Ni/MTAC.

We sincerely appreciate your insightful suggestion, which has helped us improve the clarity and reliability of our characterization. We hope this additional evidence addresses your concern and further supports our conclusions.

Comment 2: Regarding the results obtained by XPS, it is surprising that the presence of oxidized nickel species is not discussed in the framework of the supposed existence of the different types of particles, single atoms and nanoclusters.

Our response: We sincerely appreciate your valuable feedback, which has helped us strengthen the discussion of Ni oxidation states in our study.

Detailed revisions: Based on your suggestion, we have conducted a more detailed analysis of the XPS results, as outlined below.

X-ray photoelectron spectroscopy (XPS) analysis of x Ni/MTAC revealed the coexistence of $\text{Ni}^{\delta+}$ (852.4→853.0 eV, $0 < \delta < 2$) and Ni^{2+} (855.4→855.8 eV) species. The $\text{Ni}^{\delta+}$ species correspond to Ni nanoclusters, while the Ni^{2+} species, exhibiting a higher oxidation stage, are attributed to Ni single atoms due to the electronic structure modifications induced by their atomic dispersion (Fig. 2a and Supplementary Fig. 5).

As the Ni content increases from 0.5 to 1.5 wt%, the proportion of $\text{Ni}^{\delta+}$ species rises, accompanied by a gradual shift of its peaks to lower binding energies (Supplementary Table 3). This shift suggests a weakening of electron transfer within Ni, consistent with the transition from isolated single atoms to nanoclusters.

Additionally, the Mo 3d spectrum was deconvoluted into four distinct peaks at 227.5, 228.8, 231.9, and 235.3 eV, corresponding to metallic Mo, Mo-C ($3d_{5/2}$), Mo-C ($3d_{3/2}$), and Mo-O_x bonds, respectively (Fig. 2b). The presence of Mo-C and Mo-O bonds is further corroborated by the Raman spectra (Supplementary Fig. 6). Moreover, high-resolution XPS spectra of Ti 2p, C 1s, Al 2p, and O 1s provide additional confirmation of the layered structure of MTAC (Supplementary Figs. 7-11).

Figure 2a and b High-resolution Ni 2p and Mo 3d spectra.

Comment 3: But the conclusions of the study by XAS are especially surprising. First of all, the differences observed in the XANES spectra (Figure 2C) must be justified considering the resolution of the recorded spectrum, although this value is not

mentioned in the manuscript. And secondly, the results obtained by EXAFS are not credible at all. The mere visual inspection of the different TF lines (Figure 2E) does not correspond to the CN values obtained by mathematical fitting. The CN value = 0.32 obtained for the 0.5Ni/MTAC sample is completely absurd in view of the intensity of the peak at 2.09 corresponding to Ni-Ni neighbors (Figure 2e, green line). But what is more, even in the TF of this sample the presence of successive spheres of coordination can be clearly intuited by the presence of peaks in the interval between 3 and 5 Å (compare for example the green and red lines corresponding to the samples 0.5Ni and 1Ni, respectively). All of this is incompatible with the presence of single nickel atoms. Therefore, in my opinion, since the existence of SA Ni particles in the 0.5Ni catalyst is doubtful, the general conclusions of the work are not at all justified.

Our response 1: Thank you very much for your insightful and constructive comments. We sincerely appreciate your critical evaluation, which has helped us further refine our XAS analysis and interpretation. We agree with you that our previous XAS results seem unusual, as the valence of Ni was lower than 0 according to the position of the absorption edge of Ni, which is contradictory to XPS. To further verify the results, we repeated both XPS and XAS measurements. The repeated XPS results are in agreement with the previous ones, while that of XAS results show different regularity, confirming the reliability of XPS. While for the second-time XAS results this time, the absorption edges of three samples were all located within the Ni foil and NiO, i.e., the repeated XAS results now align with the XPS findings, implying the positive charge of Ni. Therefore, the previous inconsistency has been resolved.

Detailed revision 1: Based on your comments, we have intensively revised the previous descriptions and updated the related figures, as shown below.

The electronic structure and coordination environment of Ni and Mo species was further determined by X-ray absorption spectroscopy (XAS) measurements. The normalized X-ray near-edge absorption spectra (XANES) of Ni K-edge are presented in Fig. 2c. The absorption edges of three samples were all located between Ni foil and NiO, suggesting the positive charge of the Ni atoms, which is in agreement with the XPS results³⁴. Besides, the adsorption edge of 1.5Ni/MTAC shows the smallest shift to

higher energy of the three catalysts, indicative of the lowest electron density of interfacial Ni atom. As shown in normalized XANES spectra of Mo K-edge (Fig. 2d), the absorption edge of all samples is located between the Mo foil and MoO₂, suggesting the positively charged state of Mo, which is in good agreement with the XPS results.

Fig. 2 Electronic state and atomic structure characterization. **a** High-resolution Ni 2p spectra and **b** Mo 3d spectra. **c** The normalized Ni K-edge XANES spectra. **d** The normalized Mo K-edge XANES spectra. **e** Corresponding Fourier-transformed R-space spectra (solid lines) and fits (open points) of Fig. 2c. **f** Corresponding Fourier-transformed R-space spectra of Fig. 2d.

Our response 2: For your second question, we sincerely apologize for the previous problems. In this revision, we have re-fitted the data based on new experimental measurements. Furthermore, following the first reviewer's suggestion, we have incorporated both the Ni-Mo path and Ni-Ni path in our fitting analysis, considering the coexistence of single atoms and clusters. We appreciate your careful review and constructive feedback.

In addition, we understand your concerns regarding the obtained CN values and their consistency with the raw EXAFS spectra. Indeed, the Ni-Ni coordination number appears relatively high, which may seem contradictory to the presence of single-atom

Ni species. However, it is important to note that our sample is not composed entirely of isolated single atoms. As we reported earlier, statistical analysis indicates that in the 0.5Ni/MTAC sample, 89% of Ni species exist as single atoms, while the remaining fraction forms nanoclusters.

From an active site perspective, single atoms constitute the majority of the catalytically active sites. However, when considering the total number of Ni atoms, nanoclusters contain multiple Ni atoms, meaning that the overall contribution of Ni-Ni bonding is more pronounced. Since synchrotron-based XAS is a bulk characterization technique, the obtained CN values reflect the averaged electronic and structural state of all Ni species in the sample. This explains why the Ni-Ni coordination number appears higher than expected for a purely single-atom system, while still being consistent with the observed single-atom fraction.

Detailed revision 2: The detailed revisions as well as the updated figures are provided below.

The coordination environments were further analyzed by the extended X-ray absorption fine structure (EXAFS) at the K-edge of Ni and Mo. The Ni K-edge EXAFS of Ni foil shows a main peak at 2.09 Å, which was attributed to the Ni-Ni scattering path (Fig. 2e and Supplementary Figs. 14, 15). With the decrease in the Ni loadings, the peak gradually shifted to longer R value, owing to the formation of Ni-Mo path. Compared to Ni foil, the intensity of the Ni-Ni scattering path gradually diminished, confirming the interaction between Ni and MTAC support with the formation of Ni-Mo path, corresponding to coordination number reduction from 12.0 to 9.6, 8.5, and 6.7 (Supplementary Table 4). The lengths of Ni-Ni/Mo bonds in 0.5, 1, and 1.5Ni/MTAC further increased compared to Ni foil, testifying the formation of Ni-Mo bonds. Moreover, with the increase of Ni content, the coordination number of Ni-Mo decreases, resulting in the weakening of metal-support interaction.

For Mo-edge EXAFS, all the Ni/MTAC samples exhibited strong Mo-C, Mo-Ni, and Mo-Mo scattering (Fig. 2f and Supplementary Figs. 16, 17)³⁵. The 2.27 Å contribution (not phase-corrected) was attributed to the Mo-Ni scattering, which is slightly shifted compared with the Mo-Mo scattering in Mo foil (Supplementary Table

5)³⁶. To precisely clarify the atomic dispersion and coordination conditions of Ni and Mo, the wavelet transform of EXAFS spectra were analysed (Supplementary Figs. 18, 19), further confirming the formation of the Ni-Mo coordination.

Supplementary Figure 14. XAS analysis of as-prepared samples. **a** Fourier transform of Ni K-edge EXAFS spectra in K-space. The fitting results of Ni K-edge Fourier-transform EXAFS spectra on K-space of **b** 0.5Ni/MTAC, **c** 1Ni/MTAC, and **d** 1.5Ni/MTAC.

Supplementary Figure 15. The fitting results of Ni k-edge Fourier-transform EXAFS spectra on R-space of **a** 0.5Ni/MTAC, **b** 1Ni/MTAC, and **c** 1.5Ni/MTAC.

Supplementary Figure 16. XAS analysis of as-prepared samples. **a** Fourier transform of Mo K-edge EXAFS spectra in K-space. The fitting results of Mo K-edge Fourier-transform EXAFS spectra on K-space of **b** 0.5Ni/MTAC, **c** 1Ni/MTAC, and **d** 1.5Ni/MTAC.

Supplementary Figure 17. The fitting results of Mo k-edge Fourier-transform EXAFS spectra on R-space of **a** 0.5Ni/MTAC, **b** 1Ni/MTAC, and **c** 1.5Ni/MTAC.

Supplementary Figure 18. Wavelet transformed EXAFS spectra. Ni K-edge EXAFS spectra for **a** Ni foil, **b** 0.5Ni/MTAC, **c** 1Ni/MTAC, and **d** 1.5Ni/MTAC.

Supplementary Table 4. Ni K-edge EXAFS fitted parameters for as-prepared catalysts.

Samples	shell	CN ^a	R(Å) ^b	$\sigma^2 \times 10^2$ (Å ²) ^c	ΔE_0 (eV) ^d	R factor ^e
Ni foil	Ni-Ni	12.0*	2.482	0.61	7.6	0.0008
NiO	Ni-O	5.9	2.078	0.51	-3.3	0.0055
	Ni-Ni	12.2	2.949	0.59		
0.5Ni/MTAC	Ni-Ni	6.7	2.477	0.82	2.3	0.0051
	Ni-Mo	2.3	2.720			
1Ni/MTAC	Ni-Ni	8.5	2.489	0.72	6.0	0.0090
	Ni-Mo	1.3	2.790			
1.5Ni/MTAC	Ni-Ni	9.6	2.490	0.81	4.1	0.0052
	Ni-Mo	0.4	2.790			

^a Coordination numbers.

^b Bond distance.

^c Debye-Waller disorder factor.

^d adsorption edge offset.

^e goodness of fit.

* This value was fixed during EXAFS fitting, based on the known structure of Ni.

Supplementary Figure 19. Wavelet transformed EXAFS spectra. Mo K-edge EXAFS spectra for **a** Mo₂C. **b** 0.5Ni/MTAC. **c** 1Ni/MTAC, and **d** 1.5Ni/MTAC.

Supplementary Table 5. Mo K-edge EXAFS fitted parameters for as-prepared 0.5, 1, and 1.5Ni/MTAC catalysts.

Samples	shell	CN ^a	R(Å) ^b	$\sigma^2 \times 10^2$ (Å ²) ^c	ΔE_0 (eV) ^d	R factor ^e
Mo foil	Mo-Mo	8.0*	2.717	0.38	5.3	0.0013
	Mo-Mo	6.0*	3.130	0.34		
Mo ₂ C	Mo-C	2.7	2.105	0.40	-7.6	0.0148
	Mo-Mo	6.4	2.982	0.75		
0.5Ni/MTAC	Mo-C	3.0	2.083	0.38	-14.5	0.0145
	Mo-Ni	4.4	2.788	0.71		
	Mo-Mo	6.5	3.237	0.72		
1Ni/MTAC	Mo-C	3.0	2.085	0.43	-13.4	0.0175
	Mo-Ni	4.3	2.788	0.71		
	Mo-Mo	6.3	3.237	0.71		
1.5Ni/MTAC	Mo-C	2.9	2.073	0.34	-12.9	0.0146
	Mo-Ni	4.0	2.789	0.68		
	Mo-Mo	5.8	3.237	0.68		

^a Coordination numbers.

^b Bond distance.

^c Debye-Waller disorder factor.

^d adsorption edge offset.

^e goodness of fit.

* This value was fixed during EXAFS fitting, based on the known structure of Mo.

A point-to-point response to the reviewer's comments

Reviewer #1:

I'm satisfied with the authors' response, and the current version can be accepted.

Our response: Thank you for your feedback and approval. We appreciate your time and consideration!

Reviewer #2:

Required corrections were made, so paper now can be accepted. Since it is revised version, it is not essential to repeat that all is proper, interesting for readers and significant.

Our response: Thank you for your valuable feedback and for accepting the revised version of the paper. We appreciate your time and consideration!

Reviewer #3: Thank you for the answers to my remarks. In my opinion, the authors partially resolve the issues, but serious doubts remain about the main conclusions of the article.

Our response: Thank you very much for your valuable feedback and constructive comments. We appreciate your thoughtful assessment of our manuscript and the opportunity to improve its clarity and scientific rigor. Based on your insightful comments and valuable advice, we have responded all the comments and carefully revised the manuscript accordingly.

Specific response:

Comment 1: Regarding the XPS results, the authors now mention and discuss the existence of Ni²⁺ species, and they directly assign it to Ni single atoms. But this assignment must be further demonstrated. Also, they claim that the observed shift of the signal of reduced Ni species, assign to nanoclusters, is due to the transition from

isolated single atoms to nanoclusters. I cannot see how this transition can modify the BE of this signal.

Our response: Thank you for the reviewer's comments. Regarding XPS results, several literature reported can support the assignment of Ni²⁺ species to Ni single atoms and Ni^{δ+} species to Ni nanoclusters. For example, Guo *et al.* studied the role of Ni single atoms supported on Mo₂C in hydrodeoxygenation reactions, where they observed and discussed the presence of Ni²⁺ species, which aligns with our findings (Guo H. et al. Mechanistic Insights into Hydrodeoxygenation of Lignin Derivatives over Ni Single Atoms Supported on Mo₂C. ACS Catal., 2024, 14: 703-717). Additionally, Zhang *et al.* investigated single-atom nickel on MoS₂ surfaces for hydrogen evolution, providing further evidence for the existence and distribution of Ni²⁺ species (Zhang X. et al. Single-atom nickel anchored on surface of molybdenum disulfide for efficient hydrogen evolution. J. Electroanal Chem., 2021, 894: 115359). These references support our discussion of the relationship between Ni²⁺ and Ni single atoms.

Furthermore, as the Ni content increases from 0.5 wt% to 1.5 wt%, we observe a decrease in the proportion of Ni²⁺. This trend is consistent with the reduction in Ni single atoms and the increase in Ni nanoclusters as the Ni content increases, which correlates well with the changes observed in Ni^{δ+} and Ni²⁺ species. Therefore, we conclude that the Ni²⁺ species is attributed to the existence of Ni single atoms.

Regarding the comment "Also, they claim that the observed shift of the signal of reduced Ni species, assign to nanoclusters, is due to the transition from isolated single atoms to nanoclusters. I cannot see how this transition can modify the BE of this signal.". From Supplementary Table 3, we can see that the Ni^{δ+}/Ni²⁺ ratios increase with the Ni contents, and from Figure 2a, the binding energies of Ni nanoclusters also increase with the decrease of Ni introduction. Based on your suggestion, we provided a more clear analysis: As the Ni content decreases from 1.5 to 0.5 wt%, the proportion of Ni^{δ+} species reduces, accompanied by a gradual shift of its peaks to higher binding energies (Supplementary Table 3). The shift of the Ni^{δ+} peak towards higher binding energies with Ni contents is possibly attributed to the reduction in cluster size, which

leads to a change in its coordination environment, resulting in a decrease in the electron cloud density around the Ni clusters.

Comment 2: Regarding the results obtained through XAS, I do not understand the meaning of the phrase “while that of XAS results show different regularity” (page 30). What was wrong with the first measurements?

Our response: Thank you for your valuable comment. Regarding our response "while that of XAS results show different regularity", the initial synchrotron data might have been affected by instrumental errors, which caused the Ni valence to appear lower than zero, which is obviously inconsistent with conventional understanding. After conducting a second-time measurements, we found that the absorption edges of three samples were all located between Ni foil and NiO, which is in agreement with other characterization results, confirming the validity of the second set of XAS data. We hope this clarification addresses your concern.

Comment 3: XANES included in Figure 2a show similar trends for the 3 samples, indicating no important differences between the structure of Ni particles. Thus, these result do not allow to conclude about the nature of the supposed single atoms. The TFs in Figure 2e point in the same direction. They do not indicate or rule out the existence of single atoms. Finally, the discrepancies between the experimental signals and the fitting included in Figure S14 seem to be similar o even higher to the Ni-Mo contribution. Therefore, it is difficult to draw any conclusions about this kind of interaction.

Our response: Thank you for your valuable comment. We appreciate your insights, and we would like to address the points you raised: i) XANES results: The absorption edges of the three samples, as shown in Figure 2a, are located between Ni foil and NiO, specifically following the sequence of $0.5\text{Ni/MTAC} > 1\text{Ni/MTAC} > 1.5\text{Ni/MTAC}$. This trend demonstrates differences in the electronic states of the samples, which indicates subtle variations in the electronic environment of Ni atoms across the samples; ii) Ni coordination number: The reduction in the Ni coordination number indirectly

suggests an increase in the number of isolated Ni single atoms as the Ni loading decreases. This supports the notion that lower Ni content leads to a higher proportion of single atoms, which is consistent with our hypothesis and conclusions; iii) HR-TEM results: The HR-TEM results further confirm the presence of isolated Ni single atoms, providing direct evidence for their existence. This observation strengthens the argument that Ni single atoms are indeed present in the samples, despite the similar XANES trends.

Regarding the TFs in Figure 2e, we agree that they do not conclusively indicate or rule out the existence of single atoms. However, when combined with the XANES, coordination number data, and HR-TEM results, we are confident in the presence of Ni single atoms in the samples.

Regarding your final comment, we agree with you that the discrepancies between the experimental signals and the fitting included in Figure S14 seem to be similar or even higher to the Ni-Mo contribution. However, our electron microscopy and XPS results suggest the presence of Ni-Mo interactions. Therefore, we included the Ni-Mo shell in the fitting process, and the resulting fit was better compared to using only a Ni-Ni shell. This further supports the potential existence of Ni-Mo interactions.

We hope this response clarifies the points raised, and we appreciate your thorough review.